# ADS THAT STICK: NEAR-OPTIMAL AD OPTIMIZATION THROUGH PSYCHOLOGICAL BEHAVIOR MODELS

**Kailash Gopal Darmasubramanian**
Department of Computer Science and Engineering
Indian Institute of Technology, Madras
cs22b098@smail.iitm.ac.in

**Akash Pareek**
Department of Computer Science and Automation
Indian Institute of Science
Pareek.akash.cs@gmail.com

**Arindam Khan**
Department of Computer Science and Automation
Indian Institute of Science
arindamkhan@iisc.ac.in

**Arpit Agarwal**
Department of Computer Science and Engineering
Indian Institute of Technology, Bombay
agarpit@outlook.com

## ABSTRACT

Optimizing the timing and frequency of advertisements (ads) is a central problem in digital advertising, with significant economic consequences. Existing scheduling policies rely on simple heuristics, such as uniform spacing and frequency caps, that overlook long-term user interest. However, it is well-known that users' long-term interest and engagement result from the interplay of several psychological effects (Curmei, Haupt, Recht, and Hadfield-Menell, ACM CRS, 2022).

In this work, we model change in user interest upon showing ads based on three key psychological principles: *mere exposure*, *hedonic adaptation*, and *operant conditioning*. The first two effects are modeled using a concave function of user interest with repeated exposure, while the third effect is modeled using a *temporal decay* function, which explains the decline in user interest due to overexposure. Under our psychological behavior model, we ask the following question: Given a continuous time interval $T$, how many ads should be shown, and at what times, to maximize the user interest towards the ads?

Towards answering this question, we first show that, if the number of displayed ads is fixed, then the optimal ad-schedule only depends on the operant conditioning function. Our main result is a quasi-linear time algorithm that outputs a *near-optimal* ad-schedule, i.e., the difference in the performance of our schedule and the optimal schedule is exponentially small. Our algorithm leads to significant insights about optimal ad placement and shows that simple heuristics such as uniform spacing are sub-optimal under many natural settings. The optimal number of ads to display, which also depends on the mere exposure and hedonistic adaptation functions, can be found through a simple linear search given the above algorithm. We further support our findings with experimental results, demonstrating that our strategy outperforms various baselines.

## 1 INTRODUCTION

Digital advertising forms the backbone of today's trillion-dollar internet economy, serving as a primary channel for both acquiring new customers and sustaining engagement with existing ones. Since user attention is scarce and capturing it carries significant economic value, *optimizing the timing and frequency of ads* becomes a critical challenge for advertisers. This issue arises across many settings, including placing ads in (live) video or audio streams, sending push notifications to app users, or embedding sponsored content within or across user sessions. In each case, the objective is to maximize user engagement and recall while mitigating long-term *fatigue or satiation*.

A long line of empirical work in behavioral psychology has shown that the temporal spacing and frequency of ads have a significant effect on the memory retention and fatigue of the customer (Singh et al., 1994; Sahni, 2015; Curmei et al., 2022). For instance, following an initial positive neural

response to repeated stimuli, individuals tend to revert toward a baseline level of interest, causing the "thrill" of the same message to fade. This initial boost is known as *mere exposure*, while the subsequent tapering-off is termed *hedonic adaptation* in the behavioral psychology literature (Curmei et al., 2022). Moreover, insufficient spacing between two ads can drain attention and affect memory retention in the long run (Singh et al., 1994). This effect is referred to as *operant conditioning*.

Despite the importance of this problem, there has been relatively little work studying the long-term cognitive and psychological effects of ad impressions (Sahni, 2015; Aravindakshan & Naik, 2015; Rafieian, 2023) (see Section 1.1 for more details). Most existing approaches for ad placement rely on simple heuristics such as uniform spacing, front-loading, or frequency caps (Aravindakshan & Naik, 2015; Rafieian, 2023; Despotakis et al., 2021), or on short-sighted policies that treat each ad impression as independent of the others (Theocharous et al., 2015).

Recent empirical studies have tried to move beyond these simple heuristics. For instance, (Freeman et al., 2022) ran an experiment with 327 participants to test where ads should be placed in order to reduce negative reactions such as *anger, irritation, and perceived intrusiveness.* Their main hypothesis was that *"Mid-roll ads will elicit more anger and be seen as more intrusive than preroll ads"*, and their experiments confirmed this. They also tested several other hypotheses, all arriving at the same conclusion that placing ads at the beginning is generally more effective than inserting them in the middle. Similarly, (Ritter & Cho, 2009) conducted an experiment with 129 participants on audio podcasts. Their experiments supported the hypothesis that *"Advertising at the beginning of podcasts will generate less intrusiveness, less irritation, more favorable attitudes toward an ad, and less ad avoidance than advertising in the middle of podcasts".* Other studies also suggest similar trends. For example, (Goldstein et al., 2011) argues that a mix of ad placements, at the beginning, at the end, and a small number evenly spaced in the middle, can create a more positive overall experience for users.

These studies motivate us to ask whether there is an underlying psychological reward model that can provide an explanation for these empirical findings:

**Question 1.1.** *Can we design a theoretical reward model for ad scheduling that captures the users' psychological behavior?*

Ideally, the optimal policy under this reward model should be consistent with the empirical observations cited above. Moreover, this model should have tunable parameters so that it can adapt to different real-world settings. This further motivates an algorithmic question about computing the optimal ad schedule under various parameter settings:

**Question 1.2.** *Under such a reward model, can the optimal ad schedule be computed efficiently?*

To answer Theorem 1.1, we study a dynamic model of user interest under repeated ad exposures. Let $n + 1$ ads be shown at time $\bar{t} = (t_0, t_1, \ldots, t_n)$, where each $t_i \in [0, T]$ and $t_i \leq t_{i+1}$ for all $i \in \{0, 1, \ldots, n\}$. Here, $t_i$ represents the time at which the $i^{\text{th}}$ ad is shown, and without loss of generality, we set $t_0 = 0$ and $t_n = T$. We refer to $\bar{t}$ interchangeably as a *strategy* or a *schedule*. Given a time horizon $T$ and a strategy $\bar{t}$, the reward obtained from showing the $i^{\text{th}}$ ad at time $t_i$ is denoted by $R(\bar{t}, i)$. [1] The total reward associated with strategy $\bar{t}$ is then defined as $R(\bar{t}) = \sum_{i=0}^{n} R(\bar{t}, i)$.

To capture user psychology in our model, we study $R(\bar{t}, i)$ as a combination of two functions. The first function captures the (positive) mere exposure and the hedonic adaptation effects and depends only on the number of ads shown previously. We denote this function by $B : \mathbb{Z}_{\geq 0} \to \mathbb{R}_+$, a concave function with respect to the number of ads shown till now, and represents the reward when the $i^{th}$ ad is shown. The concavity of the function $B$ is justified by the 'diminishing returns' property implied by mere exposure and hedonic adaptation, and is common in recent literature that considers the dynamic effect of actions (Patil et al., 2023; Blum & Ravichandran, 2025). The second function is a temporal exponential decay function with parameter $\delta \in [0, 1]$ which captures the (negative) operant conditioning effect so that reward at time $t$ decreases by $\delta^{t-t'}$ for any ad shown previously at time $t'$. This temporal decay function is motivated by the classical theory of (Ebbinghaus, 1913) on forgetting curves, which hypothesizes the exponential decline of memory retention over time. This model is similar to the influential *goodwill stock* model of (Nerlove & Arrow, 1962), and *exponential discounting* models used in control theory (Leqi et al., 2021).

---

[1]Throughout, we use the term "reward" to describe the advertiser's optimization objective. Depending on the context, this could represent probability of purchase, user satisfaction with product, or engagement with ads.

The parameter $\delta$ plays a key role in capturing different types of user psychological behavior, such as anger, irritation, intrusiveness, or interest. A low value of $\delta$ indicates that past ad impressions have little effect on the user and therefore additional ads do not strongly reduce engagement. In contrast, a high value of $\delta$ implies that ads have more long-term effects on the user (Curmei et al., 2022). This makes a user highly sensitive when an ad is shown, which can quickly lead to irritation, loss of interest, or a perception of intrusiveness. Thus, the parameter $\delta$ provides a compact way to encode the psychological effects observed in experimental studies such as (Freeman et al., 2022; Ritter & Cho, 2009; Goldstein et al., 2011).

Given these functions, the goal is to find the number of ads $n + 1$ and the strategy $\bar{t} = (t_0, \ldots, t_n) \in [0, T]$, such that the total reward $R(\bar{t})$ is maximized. Given this model of user reward, we first show that the problem has a special structure: if the number of ads $n + 1$ is fixed, then the optimal display timing only depends on the function capturing operant conditioning.

To answer Theorem 1.2, our main algorithmic result is a quasi-linear time algorithm that, given a fixed number of ads $n + 1$, outputs a *near-optimal* ad-schedule. Let $t_i^*$ denote the optimal time to place the $i$-th ad. We show that each $t_i^*$ can be approximated by $t_i$ (produced by our algorithm) with exponentially small error, namely, $t_i^* - \frac{1}{2^n} \leq t_i \leq t_i^* + \frac{1}{2^n}$. Our result is based on several key insights: (1) the operant conditioning function is strictly convex, leading to a single global optima, (2) the global optima has a special structure such that each $t_i$ can be recursively found using $t_1, \ldots, t_{i-1}$, (3) $t_1$ can be found using binary search with *bounded error* and, (4) error propagation in the recursive computation can be controlled. Next, to determine the optimal number of ads to display, which depends on the mere exposure and hedonic adaptation functions, we perform a simple linear search, using the result discussed above.

We further support our theoretical findings using simulations where we compare the performance of different strategies under our reward model. We demonstrate that our ad placement strategy outperforms other baselines for various values of $\delta \in [0, 1]$.

**Key insights from our model and relation to previous empirical findings.** Recall that experimental studies (Freeman et al., 2022; Ritter & Cho, 2009; Goldstein et al., 2011) suggest that placing ads at the beginning and end is generally more effective than placing them in the middle. While these findings provide useful evidence regarding effective ad placement, they are limited in scope and cannot establish that such a strategy is universally optimal across all scenarios. This motivates the need for a theoretical framework that can explain and generalize these observations. In particular, our model offers deeper insights into ad placement strategies by explicitly accounting for all values of $\delta \in [0, 1]$. To systematically understand how $\delta$ influences the optimal policy and how ad placement strategies may vary across different values of $\delta$, we analyze the optimal ad policy in our model for all values of $\delta \in [0, 1]$. Our analysis demonstrates that the structure of the optimal schedule changes as $\delta$ changes. Building on this, our near-optimal ad scheduling algorithm provides not only practical scheduling strategies but also valuable theoretical insights into how ad placement should adapt across the full spectrum of $\delta$. To be precise, we observe that for small $\delta$, the ads in the optimal strategy $\bar{t}$ are placed almost evenly in $[T]$. As $\delta$ increases, more and more ads start to concentrate at $0$ and $T$ such that the first $t_i > 0$ moves towards $0$, and the last $t_j < T$ moves towards $T$. The remaining ads are evenly spaced between $t_i$ and $t_j$. All these insights show that the optimality of heuristics such as uniform spacing or placing more ads at the beginning, depend on the value of $\delta$, highlighting the need to adapt schedules based on user behavior (i.e., $\delta$).

## 1.1 RELATED WORK

**Behavioral psychology.** While the work on behavioral psychology is vast, in our work we follow (Curmei et al., 2022) and focus on three well-studied phenomena from psychology: mere exposure, operant conditioning, and hedonic adaptation. Several works in behavioral psychology has empirically shown the effect of mere exposure and hedonic adaptation under various settings (Cox & Cox, 2002; Fang et al., 2007; Chugani et al., 2015; Yang & Galak, 2015). The most relevant to our work is the study of (Hekkert et al., 2013) who found that attractiveness of a product increased with the number of times it was shown to a user. Moreover, in a similar context (Nelson & Meyvis, 2008) also found evidence of hedonic adaptation as a function of the number of exposures. Similarly, operant conditioning has also been well-studied (See (Cooper et al., 2007) and references therein). While operant conditioning might have additional connotations in the psychology literature, we mainly use it to model the 'annoyance' or 'satiation' effect of repeated exposures (Sahni, 2015). There has been

some effort on psychology-aware recommendation systems (Curmei et al., 2022; Jesse & Jannach, 2021), however, incorporating psychological effects in advertising has received less attention.

**Value of $\delta$ in real world.** There exists some work related to determining the exact curve to analyze user retention of content. In the work of (Curmei et al., 2022), the authors note that in real-world scenarios, the value of $\delta$ is 0.98. Several other works (Murre & Dros, 2015; Goldstein et al., 2011) provide motivation for setting $\delta$ between 0.7 and 0.99. This is also the regime where uniform spacing is a bad strategy as compared to the optimal ad schedule (see Section 7).

**Ad scheduling.** The problem of optimizing ad schedules has been studied across various communities such as marketing, operations research, and machine learning. One of the earliest work is due to (Nerlove & Arrow, 1962), who proposed the goodwill-stock model that treats advertising as an investment that builds a "stock" of consumer goodwill (or awareness) which then depreciates over time. Specifically, they considered a dynamical model of change in consumer goodwill given an ad impression, and cast the problem of optimizing the ad schedule as an *optimal control* problem. However, they do not consider memory/satiation effects, and the optimal policy under their model is to show most of the ads at the start of the time-horizon. (Naik et al., 1998) extended this work to consider memory effects, however, the optimal policy under their dynamical system is not interpretable. (Sahni, 2015) conducted field experiments to demonstrate that *temporal spacing* of ads has a large effect on the memory retention and satiation of the users. More recently, the problem of optimizing ad schedules has been casted as a reinforcement learning problem (Rafieian, 2023; Theocharous et al., 2015). While these algorithms tend to be general-purpose, their solutions are hard to interpret and can be difficult to execute in practice, given the complexity of ad exchanges (Despotakis et al., 2021).

**Dynamic rewards and restless bandits.** (Leqi et al., 2021) introduced a bandit framework that models user satiation using linear dynamical systems and showed that the greedy strategy is optimal when all arms have the same base reward and decay profile. More broadly, several works have explored reward structures where the current payoff depends on historical actions and decays over time (Heidari et al., 2016; Levine et al., 2017; Seznec et al., 2019). In contrast, (Kleinberg & Immorlica, 2018) studies a setting where rewards increase with time since the last pull. Related ideas appear in models where rewards evolve based on the number of pulls or the time elapsed since the last interaction (Cella & Cesa-Bianchi, 2020; Basu et al., 2019; Warlop et al., 2018; Mintz et al., 2020).

**Digital advertising.** In (Schwartz et al., 2017), the authors investigated customer acquisition through advertisements on online platforms using multi-armed bandits, and designed a policy that achieves an 8% improvement in acquisition rate. (Adany et al., 2013) addressed the problem of allocating personalized ads to users, considering each user's profile and estimated viewing capacity. (Seshadri et al., 2015) explored advertisement scheduling to meet advertisers' campaign goals while maximizing ad-sales revenue. (Dobriţa et al., 2025) proposed a framework that leverages $k$-nearest neighbors to predict ad positions, enhancing ad scheduling optimization. Aiming to maximize viewership under budget constraints, (Czerniachowska, 2019) presented a scheduling solution aligned with advertisers' budgets. (Sumita et al., 2017) developed a $(1 - \epsilon)$-competitive algorithm for envy-free allocation of video ads where $\epsilon > 0$ is a constant. We also note that digital advertising is a vast field with many practical considerations – here we summarize work that is closely related to our theoretical modeling.

### 1.2 ORGANIZATION

In Section 2, we formalize the problem, and in Section 3, we present a near-optimal algorithm for scheduling ads. Next, in Section 4, we analyze this algorithm for a special case, followed by a discussion of the broader implications of our work in Section 5. In Section 6, we give an overview of our analysis for the general case. We support our theoretical findings with experiments in Section 7, and conclude by outlining the key takeaways, limitations, and possible extensions in Section 8. All remaining details and proofs are provided in the Appendix.

## 2 PROBLEM DESCRIPTION

Let $T$ denote the total time horizon, which is known to us in advance. We consider the setting where we have to display $n + 1$ homogeneous ads in the continuous time interval $[0, T]$. We assume for now that displaying these ads is instantaneous, though in Section E, we will show how to extend this to the case when each ad needs the same amount of time. As discussed before, the function $R(\bar{t}, i)$ is composed of two parts. The first part $B(i)$ is the reward when the $i^{th}$ ad is shown (note that $i$ could

also be $0$ and corresponds to the ad shown at $t_0$), and is simply a function of the number of times we have shown the ad previously, and not of the times at which we show ads. In this work, we consider $B(i)$ to be a concave function, for example, the sigmoid function $B(i) = \frac{1}{1+e^{-ci}}$, but in general, any function would work. The second function depends on both the number of times we show ads and the times $t_i$ at which ads were shown. Let $\bar{t} = (t_0, t_1, \ldots, t_{n-1}, t_n)$ denote the time and order in which ads were displayed. The second function is $\gamma \cdot \sum_{i=0}^{l-1} \delta^{t_l - t_i}$, where $\delta \in [0, 1]$ and $\gamma \in \mathbb{R}_+$ is a constant that parameterizes the strength of this effect. The term $\sum_{i=0}^{l-1} \delta^{t_l - t_i}$, $\delta \in [0, 1]$ captures the temporal decay in the loss when the $l^{th}$ ad is shown. Hence, the reward for the $i^{\text{th}}$ ad under strategy $\bar{t}$ is given by $R(\bar{t}, i) = B(i) - \gamma \sum_{j=0}^{i-1} \delta^{t_i - t_j}$.

Let $\tilde{n} \in \mathbb{N}$ be an upper bound on the number of ads that can be shown in $[0, T]$. Our objective is to find the number of ads $n + 1 \leq \tilde{n}$, and the strategy $\bar{t} = (t_0, \ldots, t_n)$ according to which ads should be shown so that the reward $R(\bar{t})$ is maximized.

The reward $R(\bar{t})$ can be written as:

$$R(\bar{t}) = \sum_{i=0}^{n} R(\bar{t}, i) = \sum_{i=0}^{n} \left( B(i) - \gamma \sum_{j=0}^{i-1} \delta^{t_i - t_j} \right) = \left( \sum_{i=0}^{n} B(i) \right) - \left( \gamma \sum_{j < i} (\delta^{t_i - t_j}) \right).$$

Observe that the first term $\sum_i B(i)$ is independent of the times at which we show the ads, and only depends on the number of ads themselves. The coefficient of the second term, i.e., $\gamma$, only plays the role of a scaling factor. Hence, maximizing $R(\bar{t})$ for a given value of $n$ is equivalent to minimizing the *loss* $L(\bar{t})$, where $L(\bar{t})$ is defined as $L(\bar{t}) = \sum_{j < i} (\delta^{t_i - t_j})$.

Once we can find the optimal value of $R(\bar{t})$ for strategies that involve showing $n + 1$ ads, we can iterate over $\tilde{n}$ possible values to find the optimal number of ads to be shown. We provide a summary of various notations involved in Table 1. Next, we provide an overview of the algorithm to find the optimal number of ads to be shown to maximize the reward, and the optimal strategy $\bar{t}$ to show these ads.

| Description | Notation |
|---|---|
| Number of ads | $n + 1$ |
| Decay parameter | $\delta$ |
| Time Horizon | $T$ |
| Strategy | $\bar{t} = (t_0, \ldots, t_n)$ |
| Interest Function | $B(i)$ |
| Loss Function | $L(\bar{t}) = \sum_{j < i} (\delta^{t_i - t_j})$ |
| Reward Function | $R(\bar{t}) = (\sum_{i=0}^{n} B(i)) - \left( \gamma \sum_{j < i} (\delta^{t_i - t_j}) \right)$ |

Table 1: Summary of notation

## 3 NEAR-OPTIMAL ALGORITHM FOR AD SCHEDULING

In this section, we provide an overview of our near-optimal algorithm for ad scheduling. We first consider the case when we know $n + 1$, the number of ads to show. Given the time horizon $T$ and $\delta$, we want to compute the schedule $\bar{t} = (t_0, t_1, \ldots, t_n)$, such that for all $0 \leq i \leq n - 1$, $t_i \leq t_{i+1}$.

The first step in our algorithm is to compute the number of ads to show at time $0$ and $T$. Note that multiple ads can be shown at the same time, given the instantaneous nature of the ads (we relax this assumption in Section E). Let $t_a > 0$ be the first non-zero time at which an ad is shown, i.e., $a - 1$ ads are shown at time $0$. We will define the quantity $T_i$ as $T_i := \delta^{t_i}$, and throughout the paper we derive our results in terms of $T_i$ for $i \in \{0, \ldots, n\}$. Given the parameters $n, T, \delta$, our first algorithm (Algorithm 1) outputs the value of $a$ and $T_a = \delta^{t_a}$. Later in our analysis, we show that all the subsequent ad timings can be found once we know $a$ and $T_a$. We now describe the algorithm to find $a$ and $T_a$.

---

**Algorithm 1** Algorithm to obtain $a$ and $T_a$

---

1: **Input:** $n, \delta, T$.
2: Find the smallest value of $a \in [n/2]$ s.t. $\frac{a^{n-2a}}{(a+1)^{n-2a}} > \delta^T$ and $\frac{1}{a^2} \cdot \frac{(a\delta^T)^{n+2-2a}}{(a\delta^T+1)^{n-2a}} < \delta^T$ holds.
3: For the above value of $a$, define the function $h(T_a) = \frac{1}{a^2} \cdot \frac{(aT_a)^{n-2a+2}}{(1+aT_a)^{n-2a}}$
4: Compute the solution for the equation $h(T_a) = \delta^T$ via binary search. For binary search, initialize the search space to be $[\delta^T, 1]$. (see Section C for more details)
5: **return** $(a, T_a)$

---

As Algorithm 1 returns $a$, we now know the first non-zero time to show an ad. This means that we show $a - 1$ ads each at $t_0 = 0$ and at time $t_n = T$. Note that for the corner case when the value of $a$ does not exist, and for a detailed algorithm, refer to Section D. Once $a$ and $T_a$ is known, Algorithm 2 shows how to find the near-optimal schedule.

---

**Algorithm 2** Algorithm to obtain near-optimal schedule

---

1: **Input:** $n, \delta, T, a, T_a$
2: Set $t_i = 0, \forall i \in \{0, \ldots, a-1\}$ and $t_j = T, \forall j \in \{n - 2a + 1, \ldots, n\}$
3: Set $t_a = \ln(T_a)/\ln(\delta)$ and $t_{n-a} = T - t_a$
4: Distribute the remaining $t_{i+a}, \forall i \in \{0, \ldots, n-2a\}$ such that $t_{i+a} = t_a + i \cdot (t_{n-a} - t_a)/(n - 2a)$

---

Algorithm 2 provides a near-optimal ad schedule when the number of ads $n$ is known in advance. If $n$ is unknown, we use Algorithm 3 to determine it. Below, we show how to compute the optimal number of ads $n$, given an upper bound $\tilde{n}$ on the total ads that can be shown within the time horizon $T$.

---

**Algorithm 3** Optimal number of ads

---

1: **Input:** $\delta, T, \tilde{n}$
2: For each $n \in [\tilde{n}]$, compute the near-optimal strategy $\tilde{t_n}$ using Algorithm 2
3: For each near-optimal strategy, compute $R(\tilde{t_n})$
4: **return** the value of $n$ that maximizes $R(\tilde{t_n})$

---

Algorithm 3 returns the optimal number of ads to show and the near-optimal strategy to display them. For clarity, we focus our analysis on the case $a = 1$, i.e., the case where only one ad is shown at time 0 and time $T$, respectively. The general case $a \neq 1$ is discussed in Section D and follows a similar approach. When $a = 1$, our goal is to show that approximating $T_1$ helps us approximate $t_1, \ldots, t_n$, resulting in a nearly optimal schedule. In the next section, we outline our approach for the $a = 1$ case, with full proofs provided in Section 4.

## 4 ANALYSIS OF OUR ALGORITHM FOR $a = 1$

In this section, we present our analysis of the near-optimal strategy for the case $a = 1$. For the case of general $a$, an overview of our analysis is presented in Section 6. The analysis is divided into three parts: Section 4.1 proves that $L$ has a unique minimum. Section 4.2 derives closed-form expressions showing that each $T_i$ can be written in terms of $T_1$. Section 4.3 first approximates $T_1$ and then $t_1$ to an additive error of $1/2^n$. Once $T_1$ and $t_1$ are approximated, the remaining $T_i$ and $t_i$ values can be computed easily. We now start our analysis by showing that $L$ has a unique minima.

### 4.1 $L$ HAS A UNIQUE MINIMA

In this section, we want to argue about the number of minima for the loss function $L$. Specifically, we want to show that $L$ is a strictly convex function, and hence, it has at most one minima. This would imply that local minima is also the global minimum (Boyd & Vandenberghe, 2004). This characterization helps us to find the minima for $L$.

For a strategy $\bar{t}$, to form a feasible solution, it must satisfy $t_i \leq t_{i+1}$ and $0 \leq t_i \leq T$. Then, the set of feasible solutions, which we denote by $\mathcal{D}$, is defined as follows: $t_i \in \mathbb{R}, 0 \leq t_i \leq T, t_i \leq t_{i+1}, t_0 = 0, t_n = T$. We first show why $t_0 = 0$ and $t_n = T$ are required to minimize $L$.

**Lemma 4.1.** *In any optimal solution $\bar{t}$ that minimizes the loss function $L$, we have $t_0 = 0$ and $t_n = T$.*

For $L = \sum_{i>j} \delta^{t_i - t_j}$, let $a_{ij} = t_i - t_j$, where $i > j$ and denote $L' = \sum_{i>j} \delta^{a_{ij}}$. The feasible solution $\mathcal{D}'$ for $L'$ is given by: $a_{ij} \in \mathbb{R} \ \ \forall i > j, a_{ij} \geq 0, a_{n0} = T, a_{ij} + a_{jk} = a_{ik}$.

Using Theorem 4.1, we only consider those $\bar{t}$ where $t_0 = 0$ and $t_n = T$. Since $t_0$ and $t_n$ are fixed, all references to $\nabla L$ hereafter are with respect to $t_1, t_2, \ldots, t_{n-1}$. To show that $L$ has a unique global minima, we divide our analysis into four parts: (1) We establish that there exists exactly one global minima in $\mathcal{D}'$ for the function $L'$, by showing that $L'$ is strictly convex and the space $\mathcal{D}'$ is compact. (2) We show a bijection between $\mathcal{D}$ and $\mathcal{D}'$. (3) We show that, according to the previous bijection, $L'$ indeed models $L$. (4) We finally show that the minima of $L'$ corresponds exactly to the minima of $L$, and vice versa. Note that the missing proof of Theorem 4.1, the technical details to prove that $L$ has a unique minimum, and the proof of the following theorem, can be found in Section A.

**Theorem 4.2.** *The function $L$ admits a unique minima in $\mathcal{D}$.*

As we have shown that $L$ has a unique minima. To find the minima, we first argue that all $T_i$'s can be expressed in terms of $T_1$.

## 4.2 DEPENDENCY OF $T_i$'S ON $T_1$

Let $T_i = \delta^{t_i}$ for $i \in \{0, 1, \ldots, n\}$. In this section, we show that $T_2, T_3 \ldots, T_n$ can be expressed in terms of $T_1$. We have that $t_0 = 0$ and $t_n = T$. Hence, $T_0 = 1$ and $T_n = \delta^T$. Further, $t_1$ would be the time at which the second ad (the first ad after the starting time $t = 0$) is shown. In correspondence to this time $t_1$, $T_1$ would be defined as $\delta^{T_1}$.

As shown in Section 4.1, we know that $L = \sum_{j>i} \delta^{t_j - t_i}$ has a unique minima and therefore our eventual goal is to find this minima. Let $H_0 = \left(\frac{1}{\delta^{t_0}}\right), H_1 = \left(\frac{1}{\delta^{t_0}} + \frac{1}{\delta^{t_1}}\right)$, similarly $H_i = \left(\frac{1}{\delta^{t_0}} + \frac{1}{\delta^{t_1}} + \cdots + \frac{1}{\delta^{t_i}}\right) = \sum_{j=0}^{i} \frac{1}{\delta^{t_j}}$. In the following lemma, we show how $T_i$ can be expressed in terms of $T_{i-1}, H_{i-1}, H_{i-2}$, which will later help us to express $T_i$ in terms of $T_1$.

**Lemma 4.3.** *For $1 \leq i \leq n-1$, for any strategy $\bar{t} = (0, t_1, \cdots, t_{n-1}, T)$ corresponding to which $\nabla L = 0$, the following relations hold:*

$$T_i = \frac{-1 + \sqrt{1 + 4H_{i-2}H_{i-1}(T_{i-1}^2)}}{2H_{i-1}} \tag{1}$$

We now use Theorem 4.3 to show that every $T_i$ can be expressed in terms of $T_1$, which is the main theorem of this section.

**Theorem 4.4.** *For any strategy $\bar{t} = (0, t_1, \cdots, t_{n-1}, T)$ corresponding to which $\nabla L = 0$, for all $i$ in $\{1, \cdots, n-1\}$, $T_i$ can be written in terms of $T_1$ as follows:*

$$T_i = \frac{T_1^i}{(1 + T_1)^{i-1}} \tag{2}$$

Theorem 4.4 follows via a proof by induction on the expressions of $H_i$ and $T_i$, and using the induction hypothesis in Lemma 4.3. Note that in Theorem 4.4, we have shown that $T_2, \ldots, T_{n-1}$ can be expressed in terms of $T_1$. We now show how to express $T_n$ in terms of $T_1$.

**Lemma 4.5.** *For $1 \leq i \leq n$, for any strategy $\bar{t} = (0, t_1, \ldots, t_{n-1}, T)$ corresponding to which $\nabla L = 0$, the following relation holds:*

$$\frac{T_1^n}{(1 + T_1)^{n-2}} = T_n \tag{3}$$

To conclude, we derived many interesting relations between $T_1, T_2, \ldots, T_n$ in this section. The missing details and proofs can be found in Section B. In the following section, we use the theorem and lemmas of this section to find an approximate value of $T_i$ and $t_i$ for all $i \in \{1, \ldots, n-1\}$.

### 4.3 APPROXIMATING $t_i$'S

In Section 4.1, we showed that a unique minima exists for $L$, and in Section 4.2, we showed that $(T_2, \ldots, T_n)$ can be expressed in terms of $T_1$. In this section, we show how to find an approximate solution for $(T_2, \ldots, T_n)$ by finding an approximate solution for $T_1$. Note that as $T_1 = \delta^{t_1}$, once we obtain an approximate solution for $T_1$, we also get a solution for $t_1$. To this end, we describe the outline of this section before proving every detail in Section C. We first show that $T_1$ has a unique solution and then demonstrate how to approximate $T_1$ using binary search. Once we obtain an approximation for $T_1$, we use it to approximate $T_i$, $i \in \{2, \cdots, n\}$ using Theorem 4.4. Next, we approximate $t_1$ using $T_1$, which further helps in approximating $t_2, \ldots, t_n$.

For an optimal $t_1^*$, let $T_1^* = \delta^{t_1^*}$ be the optimal value of $T_i$ that satisfies $\nabla L = 0$. Similarly, define $T_2^*, \ldots, T_n^*$. First, we show the following result for $T_i$.

**Lemma 4.6.** *Assuming $T_1^*(1 - \epsilon) \leq T_1 \leq T_1^*(1 + \epsilon)$, then $T_i$ can be bounded by $T_i^* \cdot (1 - 4^n \epsilon) \leq T_i \leq T_i^* \cdot (1 + 4^n \epsilon)$.*

We now use the above lemma to approximate $t_i$, which is our main result.

**Lemma 4.7.** *For $\epsilon < \frac{1}{2}$, $t_i = \log_\delta T_i$ is bounded by $t_i^* - \frac{2\ln(2)\epsilon}{\log(1/\delta)} \leq t_i \leq t_i^* + \frac{2\ln(2)\epsilon}{\log(1/\delta)}$.*

**Corollary 4.8.** *Setting $\epsilon = \frac{1}{2^n} \cdot \frac{\log(1/\delta)}{2\ln(2)}$ we have, $t_i^* - \frac{1}{2^n} \leq t_i \leq t_i^* + \frac{1}{2^n}$.*

We have thus shown that each $t_i$ can be approximated within an exponentially small error of the optimal $t_i^*$, completing the analysis of our near-optimal strategy for $a = 1$. We next discuss several important implications and behaviors of the solution.

## 5 IMPLICATIONS

Having already established the optimality for the objective function $L$, we now proceed to analyze its structural properties. These properties reveal how the placement of advertisements varies with $\delta$, offering deeper insights into the behavior of our strategy under different values of $\delta$. We summarize here the behavior of our near-optimal solution, which is mathematically proved in Section E.

**Observation 5.1.** *The near-optimal ad schedule exhibits the following patterns as $\delta$ varies: $(a)$ As $\delta \to 0$, ads are placed at evenly spaced intervals, $(b)$ as $\delta$ increases, more ads concentrate at times $0$ and $T$, $(c)$ as $\delta$ increases, the first $t_i > 0$ moves towards 0, and the last $t_j < T$ moves towards $T$. The remaining ads are evenly spaced between $t_i$ and $t_j$.*

The above properties indicate a form of clustering behavior in our near-optimal solution. When $\delta \to 0$, it is optimal to display the advertisements at uniformly spaced intervals. As $\delta$ increases, a greater number of advertisements are placed at the endpoints, $0$ and $T$, while the remaining ones are evenly distributed in the interior of the interval. In the limit as $\delta \to 1$, the majority of the ads are concentrated at $0$ and $T$, with only a few ads placed uniformly in between.

## 6 ANALYSIS OF OUR ALGORITHM FOR GENERAL $a$

In our analysis so far, we have assumed that $T > n \log_{1/\delta}(2)$, i.e., $a = 1$. We now study the case when $T \leq n \log_{1/\delta}(2)$, i.e., $a \neq 1$. In this case, the minima do not lie in the interior of space $\mathcal{D}$. However, in Theorem 4.2, we have shown that there exists a unique minimum in $\mathcal{D}$. Therefore, such a minimum must lie on the boundary of $\mathcal{D}$.

To find the minima when $T \leq n \log_{1/\delta}(2)$, we divide our analysis into three parts. We first discuss the set of inequalities for which the minima lie at the boundary of $\mathcal{D}$. Next, similar to Section 4.2 we describe the relation between the $T_i$'s when $T \leq n \log_{1/\delta}(2)$, which will eventually help us find the minima. We provide an algorithm to iteratively check for the existence of the minimum and to find it.

In this process, we obtain the recursive relation for $T_{a+i}$

**Lemma 6.1.** *Assuming $t_0 = \cdots = t_{a-1} = 0$, we can write $T_{a+i}$ in terms of $T_a$ as*

$$T_{a+i} = \frac{1}{a} \cdot \frac{(aT_a)^{i+1}}{(aT_a + 1)^i}. \tag{4}$$

We need to add some boundary checking conditions in order to find the right value of $a$. Precisely, we need to check if at $t_a = 0$, $\frac{a^{n-2a}}{(a+1)^{n-2a}} \geq \delta^T$, at $t_a = T$, we have $\frac{1}{a^2} \cdot \frac{(a\delta^T)^{n+2-2a}}{(a\delta^T+1)^{n-2a}} \leq \delta^T$. Once we find the correct value of $a$, using similar techniques as before, we can solve for $T_a$ and recursively obtain the entire schedule. The final version of the algorithm is given in Appendix D as Algorithm 5

## 7 EXPERIMENTS

In this section, we evaluate the performance of our near-optimal strategy through four distinct experiments: (1) how does the near-optimal strategy vary as the value of $\delta$ changes? (2) comparison between our strategy and other baseline strategies? (3) how does the loss function change with a change in the number of ads? (4) how to find the optimal number of ads? We provide a brief overview of each experiment below. For detailed setup and additional results, please refer to Section F. The complete code is available at https://anonymous.4open.science/r/Ads-that-Stick-5E13/README.md.

### 7.1 VARIATION IN NEAR OPTIMAL STRATEGY WITH CHANGE IN $\delta$

In this experiment, we illustrate how our strategy evolves as the parameter $\delta$ increases from 0 to 1. Figure 1a depicts the outcome when the number of ads $n + 1$ is odd. Please refer to the Section F for the odd case. Initially, for $\delta \leq 0.4$, the ads are placed nearly equidistantly. As $\delta$ increases beyond 0.4, the ads gradually bifurcate—half of them shift towards $t = 0$, and the other half move towards $t = 20$. This experiment aligns precisely with the behavior we obtained in Theorem 5.1.

### 7.2 OUR STRATEGY VS BASELINE STRATEGIES

We compare our near-optimal strategy against three baselines - Uniform, Corner, and Random.
**Uniform:** Ads are placed uniformly over $[0, T]$, with equal spacing.
**Corner:** Half of the ads are placed at $t = 0$, half at $t = T$.
**Random:** The first and last ads are fixed at $t = 0$ and $t = T$, respectively, while the remaining $n - 2$ ads are distributed uniformly at random over the interval $(0, T)$.
Our experiment models a video streaming setting, where users engage for 1.5–2 hours and are shown approximately 15 ads. Therefore, we have $n + 1 = 15$ and $T = 100$. For this experiment, we focus on a high value of $\delta$ (greater than 0.9) as suggested in (Curmei et al., 2022). As observed in our experiment ( Figure 1b) and in Theorem E.13, when $\delta$ is small, the Uniform strategy performs well. Conversely, when $\delta$ approaches 1, the Corner strategy becomes more effective. Our results show that the near-optimal strategy adapts to $\delta$ and consistently outperforms all baseline strategies. Specifically, at $\delta$ around 0.98, our strategy outperforms all other baseline strategies at least by $10\%$.

### 7.3 CHANGE IN LOSS WITH NUMBER OF ADS

We now conduct another experiment to quantify the loss incurred by our near-optimal strategy due to the effect of operant conditioning, as the number of ads increases. Let $L^{\#}(n+1)$ denote the loss associated with showing $n$ ads under our strategy. A natural intuition is that if showing $n$ ads results in a loss of $L^{\#}(n+1)$, then doubling the number of ads would roughly double the loss, i.e., $L^{\#}(2(n+1)) \approx 2 \cdot L^{\#}(n+1)$. Our experiments actually support this intuition (see Figure 1c).

Interestingly, for smaller values of $\delta$ (around 0.7), the loss remains relatively stable even as the number of ads increases. However, we observe a sharp rise in loss as $\delta$ increases from 0.9 to 0.99, indicating increased sensitivity to operant conditioning in this regime (see Figure 1c).

### 7.4 OPTIMUM NUMBER OF ADS

In our final experiment, we demonstrate that for a given user (fixed $\delta$), the optimal number of ads $n+1$ changes under different mere exposure and hedonic adaptation functions. We use a sigmoid reward function $B(i) = k \cdot \frac{1}{1+e^{-ci}}$, where $k$ captures the overall strength of these effects and $c$ controls sensitivity to the number of ads. For $k, c > 0$, the function is concave and increasing. As shown in Figure 1d, the reward initially rises with more ads due to the dominance of mere exposure. Beyond a point, the negative impact of hedonic adaptation and operant conditioning becomes significant, causing the reward to decline. The peak of this curve corresponds to the optimal number of ads. For other related experiments, refer to Section F.

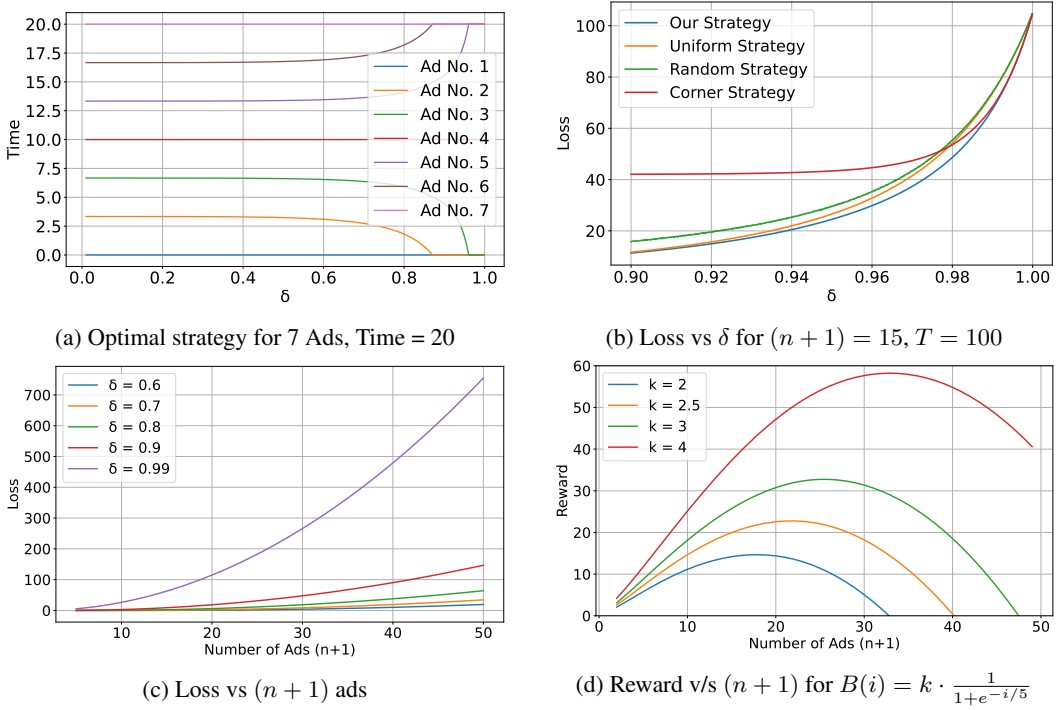

(a) Optimal strategy for 7 Ads, Time = 20

(b) Loss vs $\delta$ for $(n+1) = 15$, $T = 100$

(c) Loss vs $(n+1)$ ads

(d) Reward v/s $(n+1)$ for $B(i) = k \cdot \frac{1}{1+e^{-i/5}}$

Figure 1: (a) Change in near-optimal strategy with $\delta$ for 7 ads and $T = 20$. (b) Loss between near-optimal and baseline strategies for $n + 1 = 15$, $T = 100$. (c) Loss increases as the number of ads increases. (d) Gain function used to find the optimal number of ads for a user with $\delta = 0.9$.

## 8 CONCLUSION

In this paper, we consider a model that incorporates dynamic psychological effects – mere exposure, hedonic adaptation, and operant conditioning – into the problem of ad scheduling. We present a near-optimal strategy to schedule ads based on our behavioral model. Our strategy leads to several insights into the problem of ad scheduling, for example, equal spacing of ads might not be optimal under many settings, and it might be better to show more ads in the beginning and at the end as compared to the middle of the time-horizon. We also support these theoretical results using simulations.

### 8.1 LIMITATIONS/EXTENSIONS

**Seasonality and non-stationary rewards.** While our model can work well for real-world settings where the rewards are approximately stationary, such as inserting ads into a (live) video stream, our model does not handle scenarios where the rewards are affected by seasonality or time-of-day effect. For example, it is unlikely that sending a push notification at night will result in user attention. It will be interesting to extend our model to account for non-stationary rewards.

**Competition between ads.** In our work, we consider the optimization of the ad schedule from the point of view of a single advertiser or an ad agency running multiple homogeneous ads. In the future it will be interesting to account for externalities in the form of competing ads across various channels.

**Incorporating context or side-information.** Our model does not incorporate the context or side-information of the user or the ad into the optimization problem. This is motivated by the fact that under many scenarios, advertisers do not have access to user information, such as advertising on streaming platforms. Another future direction is to incorporate additional context of the user and ad.

**Learning the reward function.** Our current setup assumes knowledge of the reward function (including the parameter $\delta$). While our methodology is flexible enough to allow various types of reward functions, it will be interesting to study our problem as a joint learning and optimization problem in a multi-armed bandits setting. Another interesting direction would be to conduct sensitivity analysis and identify reliable indicators of $\delta$ in Ad-Data for designing experiments to estimate $\delta$.

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

## A  MISSING DETAILS OF SECTION 4.1

Before proving that $L$ has a unique minima, we want to highlight that the proofs of this section work for all values of $a$. We now provide the missing details of Section 4.1.

### A.1  PROOF OF LEMMA 4.1

We first argue that $t_0 = 0$, in any optimal solution $\bar{t}$. The argument for $t_n = T$ is quite similar.

For the sake of contradiction, assume that we have an optimal solution $\bar{t}$ where $t_0 \neq 0$. Consider a solution $t'$ where $t'_i = t_i$ for all $i > 0$ and $t'_0 = 0$.

We have

$$L(\bar{t}) = \sum_{j>0} \delta^{t_j - t_0} + \sum_{j>i>0} \delta^{t_j - t_i}$$

and

$$
\begin{aligned}
L(t') &= \sum_{j>0} \delta^{t'_j - t'_0} + \sum_{j>i>0} \delta^{t'_j - t'_i} \\
&= \sum_{j} \delta^{t_j} + \sum_{j>i>0} \delta^{t_j - t_i} \\
&< \frac{1}{\delta^{t_0}} \sum_{j} \delta^{t_j} + \sum_{j>i>0} \delta^{t_j - t_i} \\
&= \sum_{j} \delta^{t_j - t_0} + \sum_{j>i>0} \delta^{t_j - t_i} \\
&= L(\bar{t}).
\end{aligned}
$$

This implies that $L(t') < L(t)$, which is a contradiction.

### A.2  PROOF THAT $L$ HAS A UNIQUE MINIMA

As discussed in Section 4.1, our approach to proving that $L$ has a unique minima is divided into four key steps. We now prove each of these steps in detail.

**Lemma A.1.** $L'$ admits a unique minima in $\mathcal{D}'$.

*Proof.* Observe that $\delta^{a_{ij}}$ is strictly convex for constant $\delta \in (0, 1)$, as the double derivative of $\delta^{a_{ij}}$ is positive. Since the sum of strictly convex functions is also strictly convex, $L'$ is a strictly convex function.

Observe that $a_{ij} + a_{ni} + a_{j0} = a_{n0} = T$. Hence, $a_{ij} \leq T$ for all $i, j$. This means that the feasible region of $a_{ij}$ is bounded, and since the region is an intersection of linear inequalities, the feasible region of $a_{ij}$ is a polytope.

Also, note that $\mathcal{D}'$ describes a polytope and hence $\mathcal{D}'$ is a convex region. As $L'$ is strictly convex and $\mathcal{D}'$ is a convex region, $L'$ has at most one minima in $\mathcal{D}'$ (Boyd & Vandenberghe, 2004). Again, we know that polyhedra are compact sets. Therefore, the function $L'$ has a unique minimum in $\mathcal{D}'$ (Boyd & Vandenberghe, 2004). □

We now present a bijection $\mathcal{B}$ between the space $\mathcal{D}$ and $\mathcal{D}'$ and show that $\mathcal{B}$ preserves minima. Define the mapping $\mathcal{B} : \mathcal{D} \to \mathcal{D}'$ as follows:

- Given $t_i$, we set $a_{i0} = t_i$.
- Define $a_{ij} = a_{i0} - a_{j0}$.

We now show that $\mathcal{B}$ is a bijection:

$$\mathcal{B}(\bar{t})_{ij} = t_i - t_j, \forall i > j.$$

**Lemma A.2.** *The mapping $\mathcal{B}$ is a bijection from $\mathcal{D}$ to $\mathcal{D}'$.*

*Proof.* Let $\mathcal{B}(\bar{t}) = a$, for an element $\bar{t} \in \mathcal{D}$. By definition, $a_{ij} = t_i - t_j \geq 0$ since $i > j$. Also, note that $a_{ij} + a_{jk} = t_i - t_j + t_j - t_k = t_i - t_k = a_{ik}$. Hence, $a_{ij}$ satisfies the constraints associated with $\mathcal{D}'$. Hence, $\mathcal{B}$ is feasible in the domain $\mathcal{D}'$.

Observe that if $t_i \neq t'_i$, then $a_{i0} \neq a'_{i0}$, hence it is an injection.

Given $a_{ij}$, let define $\bar{t}$ where $t_i = a_{i0}$. We show that $\mathcal{B}(\bar{t}) = a$. Let $\mathcal{B}(\bar{t}) = a'$, and we will show $a' = a$. Firstly, by definition of $a'$ and $t$,

$$a'_{i0} = t_i = a_{i0}.$$

We also have

$$a'_{ij} = a'_{i0} - a'_{j0} = t_i - t_j = a_{i0} - aj0 = a_{ij}.$$

Therefore $a = a'$ and hence for each $a$, there exists a $\bar{t}$ such that $\mathcal{B}(\bar{t}) = a$. Therefore, $\mathcal{B}$ is a surjection. Since $\mathcal{B}$ is both an injection and a surjection, it is a bijection. $\square$

**Corollary A.3.** *As $\mathcal{B}$ is a bijection, it is also invertible. It can be verified that,*

$$(\mathcal{B}^{-1}(a))_i = a_{i0}.$$

To this end, we now show that $L'$ indeed models $L$ according to $\mathcal{B}$:

**Lemma A.4.** $L(t) = L'(\mathcal{B}(\bar{t}))$.

*Proof.*

$$
\begin{aligned}
L(t) &= \sum_{i>j} \delta^{t_i - t_j} \\
&= \sum_{i>j} \delta^{\mathcal{B}(t_i) - \mathcal{B}(t_j)} \\
&= \sum_{i>j} \delta^{a_i 0 - a_j 0} \\
&= \sum_{i>j} \delta^{a_{ij}} \\
&= L'(\mathcal{B}(\bar{t})).
\end{aligned}
$$

$\square$

Next, we demonstrate that $\mathcal{B}$ preserves the minima, i.e., $t^*$ is a minima of $L$ iff $\mathcal{B}(t^*)$ is a minima of $L'$.

**Lemma A.5.** $t^*$ *is a minima of $L$ iff $\mathcal{B}(t^*)$ is a minima of $L'$.*

*Proof.* Let us first prove the forward direction. If $t^*$ is a local minima of $L$, then by definition, there exists an $\epsilon$ such that for all $t, |t - t^*| < \epsilon$, so we have that $L(t) \geq L(t^*)$. Set $\epsilon' = \epsilon$. We want to show that in the neighborhood around $a^* = \mathcal{B}(t^*)$, for all $a$, $L'(a) \geq L'(a^*)$.

For the sake of contradiction, assume that there exists $a \in \mathcal{D}'$ such that $|a - a^*| \leq \epsilon'$ and $L'(a) < L'(a^*)$. Consider $t = \mathcal{B}^{-1}(a)$. First, observe that $|t - t^*| \leq \epsilon$ since $a$ has $t$ embedded in it. Also, we have $L(t) = L'(a) < L'(a^*) = L(t^*)$, which is a contradiction to the fact that $t^*$ is a local minima.

We now prove the other direction. Let $a^*$ be a local minimum for $L'$. Hence, by definition, there exists an $\epsilon$ such that for all $a, |a - a^*| < \epsilon$, we have $L(a) \geq L(a^*)$. Let us set $\epsilon' = \epsilon/2n^2$.

For the sake of contradiction assume there exists a $t \in \mathcal{D}$ such that $|t - t^*| \leq \epsilon'$ and $L(t) < L(t^*)$. Observe that $(|\mathcal{B}(t) - \mathcal{B}(t^*)|) \leq 2 \cdot \max_i(t_i - t_i^*) \cdot n^2 \leq 2n^2\epsilon'$ (using the fact that the magnitude of a vector is more than the value of each of its components). Also, $(L'(\mathcal{B}(t) - L'(\mathcal{B}(t^*)) < 0$, which is a contradiction to the fact that $a^*$ was a local minima. $\square$

Putting everything together, we now present the proof of Theorem 4.2:

## A.3 PROOF OF THEOREM 4.2

First, we showed that the function $L'$ admits a unique minima in $\mathcal{D}'$ in Lemma A.1. We then proved that the number of minima of $L$ in $\mathcal{D}$ is the same as the number of minima of $L'$ in $\mathcal{D}'$ in Lemma A.5. Hence, $L$ admits a unique minima in $\mathcal{D}$.

## B MISSING DETAILS OF SECTION 4.2

In this section, and in Appendix C, we focus on the case $a = 1$, and defer the general case ($a \neq 1$) to Appendix D. When $a = 1$, the two conditions in Line 2 of Algorithm 1 simplify to $\frac{1}{2^{n-2}} > \delta^T$ and $\frac{(\delta^T)^n}{(\delta^T+1)^{n-2}} < \delta^T$. The second condition always holds, while the first condition holds only when $T > n \log_{1/\delta}(2)$. Therefore, in this section and in Appendix C, we assume $T > n \log_{1/\delta}(2)$ whenever needed. The case $T \leq n \log_{1/\delta}(2)$, where $a \neq 1$, is handled in Appendix D. We now provide the missing details of Section 4.2. To this end, we show how $T_i$ is can be expressed in terms of $T_1, \ldots, T_{i-1}$ and $T_{i+1}$ to $T_n$.

**Lemma B.1.** *For $1 \leq i \leq n-1$, for any strategy $\bar{t} = (0, t_1, \cdots, t_{n-1}, T)$ corresponding to which $\nabla L = 0$, we have,*

$$(\delta^{t_i})^2 = \left( \frac{1}{\frac{1}{\delta^{t_0}} + \cdots + \frac{1}{\delta^{t_{i-1}}}} \right) \cdot (\delta^{t_{i+1}} + \cdots + \delta^{t_n})$$

.

*Proof.* As we know $\nabla L = 0$, therefore the partial derivative with respect to $t_1$ can be written as

$$\frac{\partial(L)}{\partial(t_1)} = 0$$

This implies that

$$\frac{\delta^{t_1}}{\delta^{t_0}} - \frac{1}{\delta^{t_1}}(\delta^{t_2} + \delta^{t_3} + \cdots + \delta^{t_n}) = 0$$

We can rewrite the above equation as

$$(\delta^{t_1})^2 = \left( \frac{1}{\frac{1}{\delta^{t_0}}} \right) \cdot (\delta^{t_2} + \delta^{t_3} + \cdots + \delta^{t_n})$$

Similarly, for any $i \in \{2, \cdots, n-1\}$, we have

$$\frac{\partial(L)}{\partial(t_i)} = 0$$

$$\delta^{t_i} \left( \frac{1}{\delta^{t_0}} + \frac{1}{\delta^{t_1}} + \cdots + \frac{1}{\delta^{t_{i-1}}} \right) - \frac{1}{\delta^{t_i}}(\delta^{t_{i+1}} + \cdots + \delta^{t_n}) = 0$$

$$(\delta^{t_i})^2 = \left( \frac{1}{\frac{1}{\delta^{t_0}} + \frac{1}{\delta^{t_1}} + \cdots + \frac{1}{\delta^{t_{i-1}}}} \right) \cdot (\delta^{t_{i+1}} + \cdots + \delta^{t_n}). \tag{5}$$

This proves the lemma. □

## B.1 PROOF OF LEMMA 4.3

As $T_i = \delta^{t_i}$, using Lemma B.1 we have

$$T_i^2 = \frac{1}{H_{i-1}} \cdot \left( \sum_{j=i+1}^{n} T_j \right) \tag{6}$$

We can rewrite the above equation as

$$H_{i-1}T_i^2 = \sum_{j=i+1}^{n} T_j \tag{7}$$

Similarly, we have

$$H_{i-2}T_{i-1}^2 = \sum_{j=i}^{n} T_j \tag{8}$$

Subtracting (8) from (7), we get

$$H_{i-1}T_i^2 - H_{i-2}T_{i-1}^2 = \sum_{j=i+1}^{n} T_j - \sum_{j=i}^{n} T_j$$

This implies

$$H_{i-1}T_i^2 + T_i - H_{i-2}T_{i-1}^2 = 0$$

Observe that we now have a quadratic equation in $T_i$. Also note that $T_i > 0$, as $\delta^{t_i} > 0$, the quadratic equation above can only have a positive root. Therefore,

$$T_i = \frac{-1 + \sqrt{1 + 4H_{i-2}H_{i-1}(T_{i-1}^2)}}{2H_{i-1}}$$

We now want to show that every $T_i$ can be expressed in terms of $T_1$. To this end, we first show that $T_2$ can be expressed as a function of $T_1$ and then formalize it for any $T_i$.

**Claim B.2.** *At any time $(t_1, \cdots, t_n)$ where $\nabla L = 0$, $T_2$ can be written in terms of $T_1$ as follows:*

$$T_2 = \frac{T_1^2}{1 + T_1}. \tag{9}$$

*Proof.* From Lemma 4.3, we have

$$T_i = \frac{-1 + \sqrt{1 + 4H_{i-2}H_{i-1}(T_{i-1}^2)}}{2H_{i-1}}$$

For $i = 2$ we get

$$T_2 = \frac{-1 + \sqrt{1 + 4H_0H_1(T_1^2)}}{2H_1}$$

As $t_0 = 0$, we have $\delta^{t_0} = 1$ which implies $H_0 = 1$ and $H_1 = \left(1 + \frac{1}{T_1}\right)$. Putting this in the above equation, we get:

$$\begin{aligned}
T_2 &= \frac{-1 + \sqrt{1 + 4 \cdot 1 \cdot (1 + \frac{1}{T_1})T_1^2}}{2 \cdot (1 + \frac{1}{T_1})} \\
&= T_1 \frac{-1 + \sqrt{4T_1^2 + 4T_1 + 1}}{2 \cdot (T_1 + 1)} \\
&= T_1 \frac{-1 + 2T_1 + 1}{2(T_1 + 1)} \\
&= \frac{T_1^2}{1 + T_1}.
\end{aligned}$$

$\square$

## B.2 PROOF OF THEOREM 4.4

We prove this theorem by induction. As our base case, we know that the theorem holds for $T_2$ (Claim B.2).

Let us assume that the induction hypothesis holds till $T_{i-1}$. We want to show that the theorem also holds for $T_i$. To start with, let us first calculate $H_k$ for $k < i$,

$$
\begin{aligned}
H_k &= \sum_{j=0}^{k} \frac{1}{T_j} \\
&= 1 + \sum_{j=1}^{k} \frac{(1+T_1)^{j-1}}{T_1^j} \qquad && \text{(Using induction hypothesis)} \\
&= 1 + \frac{1}{T_1} \cdot \frac{(\frac{1+T_1}{T_1})^k - 1}{\frac{1+T_1}{T_1} - 1} \qquad && \text{(Using geometric progression formula)} \\
&= 1 + \frac{1}{T_1} \cdot \frac{(\frac{1+T_1}{T_1})^k - 1}{\frac{1}{T_1}} \\
&= 1 + \left(\frac{1+T_1}{T_1}\right)^k - 1 \\
&= \left(\frac{1+T_1}{T_1}\right)^k.
\end{aligned}
$$

Using Equation 1, we have,

$$
\begin{aligned}
T_i &= \frac{-1 + \sqrt{1 + 4 H_{i-2} H_{i-1} (T_{i-1}^2)}}{2 H_{i-1}} \\
&= \frac{-1 + \sqrt{1 + 4 \cdot (\frac{(1+T_1)}{T_1})^{i-2} \cdot (\frac{(1+T_1)}{T_1})^{i-1} \cdot (\frac{T_1^{i-1}}{(1+T_1)^{i-2}})^2}}{2(\frac{1+T_1}{T_1})^{i-1}} \\
&= T_1^{i-1} \cdot \frac{-1 + \sqrt{1 + 4(T_1 + 1)(T_1)}}{2(1+T_1)^{i-1}} \\
&= T_1^{i-1} \cdot \frac{-1 + 2T_1 + 1}{2(1+T_1)^{i-1}} \\
&= \frac{T_1^i}{(1+T_1)^{i-1}}.
\end{aligned}
$$

**Corollary B.3.** *At any time* $(t_1, \cdots, t_n)$ *where* $\nabla L = 0$, *for all* $i$ *in* $[n-1]$, $H_i$ *can be expressed in terms of* $T_1$ *as follows:*

$$
H_i = \left(\frac{1+T_1}{T_1}\right)^i \tag{10}
$$

## B.3 PROOF OF LEMMA 4.5

From Lemma B.1, we have that

$$
T_1^2 = (T_2 + T_3 + \cdots + T_n) \tag{11}
$$

Which can be written as

$$
T_1^2 - (T_2 + T_3 + \cdots + T_{n-1}) = T_n
$$

By using Theorem 4.4, we substitute $T_i = \frac{(T_1)^i}{(1+T_1)^{i-1}}$ for $1 < i < n$, to get

$$T_2 + T_3 + \cdots + T_{n-1} = \frac{(T_1)^2}{(1 + T_1)} + \frac{(T_1)^3}{(1 + T_1)^2} + \cdots + \frac{(T_1)^{n-1}}{(1 + T_1)^n}$$

$$= \frac{(T_1)^2}{(1 + T_1)} \cdot \frac{1 - (\frac{T_1}{1+T_1})^{n-2}}{1 - \frac{T_1}{1+T_1}}$$

$$= T_1^2 - \frac{T_1^n}{(1 + T_1)^{n-2}}$$

Hence, we have

$$\frac{T_1^n}{(1 + T_1)^{n-2}} = T_n.$$

**Corollary B.4.** *For $1 \leq i \leq n$, at any time $(t_1, \cdots, t_n)$ where $\nabla L = 0$, the following relation holds:*

$$\frac{\delta^{t_1 n}}{(1 + \delta^{t_1})^{n-2}} = \delta^T. \tag{12}$$

## C  MISSING DETAILS OF SECTION 4.3

As discussed in Section 4.3, we first show that $T_1$ has a unique solution, then we approximate the $T_i$'s, and lastly we approximate $t_i$'s. We now start by showing that $T_1$ has a unique solution.

### C.1  $T_1$ HAS A UNIQUE SOLUTION

In Lemma 4.5, we showed a relation between $T_1$ and $T_n$. Here we first show that the function $\frac{T_1^n}{(1+T_1)^{n-2}}$ with respect to $T_1$ is strictly monotone and then argue that $T_1$ has a unique solution which minimizes $L$ provided that $T > n \log_{1/\delta}(2)$.

**Lemma C.1.** *Let $g(T_1) = \frac{T_1^n}{(1+T_1)^{n-2}}$. Then $g$ increases strictly monotonically with $T_1$.*

*Proof.*

$$\frac{d}{dT_1} \frac{T_1^n}{(1 + T_1)^{n-2}} = \frac{nT_1^{n-1}}{(1 + T_1)^{n-2}} - (n - 2) \frac{T_1^n}{(1 + T_1)^{n-1}}$$

$$= \frac{nT_1^{n-1} + 2T_1^n}{(1 + T_1)^{n-1}}$$

$$> 0$$

Hence, $g(T_1)$ strictly monotonically increasing with $T_1$. $\qquad\square$

Note that as $g(T_1)$ is strictly monotonically increasing with $T_1$ and $T_n = \delta^T$ is a constant, we show that $T_1$ has a unique solution. To show this, for convenience we assume that $T > n \log_{1/\delta}(2)$. We remove this assumption later in Appendix D.

**Lemma C.2.** *Assuming $T > n \log_{1/\delta}(2)$, $g(T_1) = \delta^T$ admits a unique solution.*

*Proof.* Observe that $t_1 \in \{0, \cdots, T\}$, so $T_1 \in [\delta^T, 1]$.
When $T_1 = \delta^T$,

$$g(T_1) = \frac{T_1^n}{(1 + T_1)^{n-2}}$$

$$< T_1$$

$$= T_n$$

By our assumption that $T > n \log_{1/\delta}(2)$, let us show that $\delta^T < \frac{1}{2^{n-2}}$

$$T > (n-2) \log_{1/\delta}(2)$$
$$T \log(1/\delta) > (n-2) \log(2)$$
$$\delta^T < \frac{1}{2^{n-2}}$$

When $T_1 = 1$,

$$g(T_1) = 1/2^{n-2}$$
$$> T_n \qquad \text{(Assuming } T > n \log_{1/\delta}(2)).$$

As $g(T_1) < T_n$ when $T_1 = \delta^T$ and $g(T_1) > T_n$ when $T_1 = 1$, using intermediate value theorem, there exists a point $p \in [\delta^T, 1]$ such that $g(p) = T_n$. Since $g$ is a continuous and strictly monotonically increasing function, the point $p$ is unique. Hence, $g(T_1)$ admits a unique solution. $\qquad \square$

**Lemma C.3.** *A solution to $g(T_1) = \delta^T$ corresponds to a solution to $\nabla L = 0$.*

*Proof.* We began with a set of $n$ independent equations (Equation 5) and, through algebraic manipulation, derived another set of $n$ independent equations (Equations B.4 and 4.4). Therefore, solving for $g(T_1) = \delta^T$ and computing the corresponding values of $T_i$ yields a solution that also satisfies the original equations. Since Equation 5 implies $\nabla L = 0$, this solution corresponds to a minimum of $L$.

$\qquad \square$

Lemma C.2 states that when $T > n \log_{1/\delta}(2)$, i.e., $T$ is much larger compared to $n$, there exists a unique interior point solution to our problem. In section D, we remove this assumption and show how to find the solution of $g(T_1)$. But for simplicity, we proceed with the assumption now.

## C.2 Approximating $T_1$ using Binary Search

As we know that $g(T_1)$ has a unique solution and $g$ is monotonically increasing with $T_1$, we perform binary search and estimate $T_1$ to an accuracy of $1/8^n$.

**Lemma C.4.** *Assuming $T > n \log_{1/\delta}(2)$, we estimate the value of $T_1$ to an accuracy of $1/8^n$, in polynomial time.*

*Proof.* We know, $g(T_1) = \frac{T_1^n}{(1+T_1)^{n-2}}$ is strictly monotonically increasing with $T_1$. Let

$$h(T_1) = g(T_1) - T_n.$$

As $T_n$ is a constant, $h(T_1)$ also strictly monotonically increasing with $T_1$.

Also, as $T_1 \in [\delta^T, 1]$ and $T > n \log_{1/\delta}(2)$, it can be verified that at $T_1 = \delta^T$, $h(T_1) = h(\delta^T) < 0$, and at $T_1 = 1$, $h(T_1) = h(1) > 0$.

We now perform binary search as described in Algorithm 4, to obtain $T_1$. We later show that Algorithm 4 takes $O(3n + T \log_2(1/\delta))$ iterations to obtain an approximation of $1/8^n$. Since we only do constant-time operations and function evaluations in each iteration, we need $O(\ln(nT))$ time per iteration. Therefore, the overall time to obtain an approximation of $T$ to a factor of $1/8^n$ would be $O((3n + T \log_2(1/\delta)) \ln(nT))$.

Note that the size of our search domain is initially $1 - \delta^T < 1$ and in each round, the size of our domain decreases by a factor of 2. Let $r$ be the total number of iterations, then after $r = 3n + T \log_2(1/\delta)$ iterations, our domain size decreases to a factor of less than

$$\frac{1}{2^r} = \frac{1}{2^{3n+T \log_2(1/\delta)}}$$
$$= \frac{1}{8^n} \cdot \delta^T$$
$$\leq \frac{1}{8^n} \cdot T_1 \qquad (T_1 \geq \delta^T).$$

---

**Algorithm 4** Binary Search

1: $low \leftarrow \delta^T$
2: $high \leftarrow 1$
3: $n_{rounds} \leftarrow 3n + T \log(1/\delta)$
4: $mid \leftarrow (low + high)/2$
5: $curr \leftarrow 1$
6: **while** $curr \leq n_{rounds}$ **do**
7:     $mid \leftarrow (low + high)/2$
8:     **if** $h(mid) = 0$ **then**
9:       **return** $mid$
10:     **else if** $h(mid) < 0$ **then**
11:       $low \leftarrow mid$
12:     **else**
13:       $high \leftarrow mid$
14:     **end if**
15:     $curr \leftarrow curr + 1$
16: **end while**
17: **return** $mid$

---

$\square$

For an optimal $t_1^*$, let $T_1^* = \delta^{t_1^*}$ be the optimal value of $T_i$ that satisfies $\nabla L = 0$. Similarly, define $T_2^*, \ldots, T_n^*$. In the following lemma, we bound the error in $T_i$'s with respect to the estimation we make for $T_1$.

### C.2.1    PROOF OF LEMMA 4.6

Using Theorem 4.4 we know,

$$T_i = \frac{T_1^i}{(1 + T_1)^{i-1}}$$

Let us assume that we can bound $T_1$ as

$$T_1^*(1 - \epsilon) \leq T_1 \leq T_1^*(1 + \epsilon)$$

Then we get,

$$
\begin{aligned}
T_i &= \frac{T_1^i}{(1 + T_1)^{i-1}} \\
&\leq \frac{(T_1^*(1 + \epsilon))^i}{(1 + T_1^*(1 - \epsilon))^{i-1}} \\
&\leq \frac{(T_1^*(1 + \epsilon))^i}{((1 + T_1^*)(1 - \epsilon))^{i-1}} \\
&= \frac{(T_1^*)^i}{(1 + T_1^*)^{i-1}} \cdot \frac{(1 + \epsilon)^i}{(1 - \epsilon)^{i-1}} \\
&= T_i^* \cdot \frac{(1 + \epsilon)^i}{(1 - \epsilon)^{i-1}}
\end{aligned}
$$

Using some known inequalities like $(1 + x)^r \leq 1 + (2^r - 1)x$ and $\frac{1}{1-x} \leq 1 + 2x$ for $\epsilon < 1/2$.

We have,

$$T_i \leq T_i^* \cdot \frac{(1+\epsilon)^i}{(1-\epsilon)^{i-1}}$$
$$\leq T_i^* \cdot (1+\epsilon)^i(1+2\epsilon)^{i-1}$$
$$\leq T_i^* \cdot (1+2\epsilon)^{2i-1}$$
$$\leq T_i^* \cdot (1+(2^{2i-1}-1)2\epsilon)$$
$$\leq T_i^* \cdot (1+(2^{2i}-1)\epsilon)$$

Similarly, we get

$$T_i = \frac{T_1^i}{(1+T_1)^{i-1}}$$
$$\geq \frac{(T_1^*(1+\epsilon))^i}{(1+T_1^*(1+\epsilon))^{i-1}}$$
$$\geq \frac{(T_1^*(1+\epsilon))^i}{((1+T_1^*)(1+\epsilon))^{i-1}}$$
$$= \frac{(T_1^*)^i}{(1+T_1^*)^{i-1}} \cdot \frac{(1-\epsilon)^i}{(1+\epsilon)^{i-1}}$$
$$= T_i^* \cdot \frac{(1-\epsilon)^i}{(1+\epsilon)^{i-1}}$$

Again, we know that $(1-x)^r \geq 1-(2^r-1)x$ and $\frac{1}{1+x} \geq 1-x$.

We have,

$$T_i \geq T_i^* \cdot \frac{(1-\epsilon)^i}{(1+\epsilon)^{i-1}}$$
$$\geq T_i^* \cdot (1-\epsilon)^i(1-\epsilon)^{i-1}$$
$$= T_i^* \cdot (1-\epsilon)^{2i-1}$$
$$\geq T_i^* \cdot (1-(2^{2i-1}-1)\epsilon)$$

Putting everything together, we get

$$T_i^* \cdot (1-(2^{2i}-1)\epsilon) \leq T_i \leq T_i^* \cdot (1+(2^{2i-1}-1)\epsilon)$$

We can rewrite the above equation as

$$T_i^* \cdot (1-4^i\epsilon) \leq T_i \leq T_i^* \cdot (1+4^i\epsilon)$$

Since $i \leq n$

$$T_i^* \cdot (1-4^n\epsilon) \leq T_i \leq T_i^* \cdot (1+4^n\epsilon)$$

Hence, we have proved the lemma.

**Corollary C.5.** *Setting $\epsilon = 1/8^n$, we get,*

$$T_i^* \cdot (1-1/2^n) \leq T_i \leq T_i^* \cdot (1+1/2^n).$$

The above corollary is obtained by putting $\epsilon = 1/8^n$ because we have already shown in Lemma C.4 that $T_1$ can be estimated to a factor of $1/8^n$ using Algorithm 4.

### C.3  APPROXIMATING $t_i$ USING $T_i$

As we have shown that we can estimate $T_1, \ldots, T_n$ in terms of multiplicative error, and since $T_i = \delta^{t_i}$, we now bound $t_i$ in terms of additive error.

### C.3.1 PROOF OF LEMMA 4.7

As $T_i$ satisfies

$$T_i^*(1 - \epsilon) \le T_i \le T_i^*(1 + \epsilon)$$

We have,

$$
\begin{aligned}
T_i &\le T_i^*(1 + \epsilon) \\
\delta^{t_i} &\le \delta^{t_i^*}(1 + \epsilon) \\
t_i &\ge t_i^* + \log_\delta(1 + \epsilon) \\
t_i &\ge t_i^* - \frac{1}{\log 1/\delta} \cdot \log(1 + \epsilon) \\
t_i &\ge t_i^* - \frac{1}{\log 1/\delta} \cdot (\epsilon)
\end{aligned}
$$

Similarly, we have

$$
\begin{aligned}
T_i &\ge T_i^*(1 - \epsilon) \\
\delta^{t_i} &\ge \delta^{t_i^*}(1 - \epsilon) \\
t_i &\le t_i^* + \log_\delta(1 - \epsilon) \\
t_i &\le t_i^* - \frac{1}{\log(1/\delta)} \log(1 - \epsilon) \\
t_i &\le t_i^* - \frac{1}{\log(1/\delta)} (-2 \ln(2)\epsilon) \\
t_i &\le t_i^* + \frac{2 \ln(2)\epsilon}{\log(1/\delta)}
\end{aligned}
$$

Hence, $t_i^* - \frac{2 \ln(2)\epsilon}{\log(1/\delta)} \le t_i \le t_i^* + \frac{2 \ln(2)\epsilon}{\log(1/\delta)}$.

**Corollary C.6.** *Setting* $\epsilon = \frac{1}{2^n} \cdot \frac{\log(1/\delta)}{2 \ln(2)}$ *we have,*

$$t_i^* - \frac{1}{2^n} \le t_i \le t_i^* + \frac{1}{2^n}.$$

## D ANALYSIS WHEN $a \ne 1$

In our analysis so far, we have assumed that $T > n \log_{1/\delta}(2)$, i.e., $a = 1$. We now study the case when $T \le n \log_{1/\delta}(2)$, i.e., $a \ne 1$. In this case, the minima do not lie in the interior of space $\mathcal{D}$. However, in Theorem 4.2, we have shown that there exists a unique minimum in $\mathcal{D}$. Therefore, such a minimum must lie on the boundary of $\mathcal{D}$.

To find the minima when $T \le n \log_{1/\delta}(2)$, we divide our analysis into three parts.

1. We first discuss the set of inequalities for which the minima lie at the boundary of $\mathcal{D}$.

2. Next, similar to Section 4.2 we describe the relation between the $T_i$'s when $T \le n \log_{1/\delta}(2)$, which will eventually help us find the minima.

3. We then provide an algorithm to iteratively check the existence of the minima, if it exists, and how to find it.

### D.1 FINDING THE RIGHT SUBSPACE TO SEARCH FOR THE MINIMA

This section focuses on determining the value of $a$. In our algorithm, we perform a linear search over possible values of $a$, and in each iteration, we check whether the conditions

$$\frac{a^{n-2a}}{(a+1)^{n-2a}} \ge \delta^T \quad \text{and} \quad \frac{1}{a^2} \cdot \frac{(a\delta^T)^{n+2-2a}}{(a\delta^T + 1)^{n-2a}} \le \delta^T$$

are satisfied. Theorem D.3 shows that this linear search correctly identifies the value of $a$, i.e., if $a$ satisfies these conditions, then in the optimal schedule, we have $t_i = 0$ and $t_{n-i} = T$ for all $i < a$.

From the definition of space $\mathcal{D}$ in Section 4.1, we note that the boundaries of $\mathcal{D}$ are $t_i \geq 0$, $t_i \leq T$ and $t_i \leq t_{i+1}$. But the set of equations obtained so far indicates that for the optimal $\bar{t}$, $t_i < t_{i+1}$. We therefore show that any solution with the added constraint $t_i = t_{i+1}$ also corresponds to a solution to our initial set of equations for $\mathcal{D}$. To this end, with the additional constraint that $t_i = t_{i+1}$, we obtain the following two sets of equations.

1. Substituting $t_{i+1}$ for $t_i$ in $L$ and writing $\nabla L = 0$ where $\nabla$ is taken with respect to $\{t_1, \ldots, t_{i-1}, t_{i+1}, \ldots, t_{n-1}\}$.

2. Substituting $t_i$ for $t_{i+1}$ in $L$ and writing $\nabla L = 0$ where $\nabla$ is taken with respect to $\{t_1, \ldots, t_i, t_{i+2}, \ldots, t_{n-1}\}$.

We denote $\{t_1, \ldots, t_{i-1}, t_{i+1}, \ldots, t_{n-1}\}$ as our first equation and $\{t_1, \ldots, t_i, t_{i+2}, \ldots, t_{n-1}\}$ as our second equation. We now show the following lemma.

**Lemma D.1.** *Any solution to the set of equations obtained by solving $\nabla L = 0$, under the additional assumption $t_i = t_{i+1}$, is also a solution to the original set of equations obtained by solving $\nabla L = 0$.*

*Proof.* We here present a proof sketch as the proof is similar to the proof of Lemma B.1.

Let us denote our original set of equations obtained by $\partial(L)/\partial t_j$ in Lemma B.1 by $\mathcal{E}_j$. When we perform $\partial L/\partial t_j$, in our first set of equations, it looks like we have substituted $t_i$ with $t_{i+1}$ in our set of original equations. Denote by $\mathcal{E}_j^1$ the equation obtained by $\partial(L)/\partial t_j$ in our first set of equations. For example, when $\partial L/\partial t_1 = 0$, we get

$$\frac{\partial(L)}{\partial t_1} = \delta^{t_1}(\frac{1}{\delta^{t_0}}) + (\frac{1}{\delta^{t_1}})(\delta^{t_2} + \cdots + 2\delta^{t_{i+1}} + \cdots + \delta^{t_n}).$$

Similarly, our second set of equations looks like we have substituted $t_{i+1}$ with $t_i$ in our set of original equations. Denote by $\mathcal{E}_j^2$ the equation obtained by $\partial(L)/\partial t_j$ in our second set of equations. For example, when $\partial L/\partial t_1 = 0$, we get

$$\frac{\partial(L)}{\partial t_1} = \delta^{t_1}(\frac{1}{\delta^{t_0}}) + (\frac{1}{\delta^{t_1}})(\delta^{t_2} + \cdots + 2\delta^{t_i} + \cdots + \delta^{t_n}).$$

Observe that since there is a unique minima for $L$, solving the above two equations must give the same solution (or it can be shown that both equations have no solution). We can now construct $\bar{t}' = (t_1, \cdots, t_i, t_i, \ldots, t_{n-1})$, where $\bar{t}' = (t_1, \ldots, t_{n-1})$ is the solution obtained by the first set of equations by calculating $\nabla L = 0$, once substituting $t_{i+1}$ by $t_i$.

Let us first talk about the constraints that are obtained by equation $\mathcal{E}_j^1$, where $j \neq i+1$. In our solution $\bar{t}'$, we have that $t_i = t_{i+1}$. But $\mathcal{E}_j^1$ is identical to what we obtain when we set $t_i = t_{i+1}$ in $\mathcal{E}_j$. Hence, if we start with a solution $\bar{t}'$ of $\mathcal{E}_j^1$, it will also be a solution to $\mathcal{E}_j$.

Let $\bar{t}'' = (t_1, \cdots, t_{i+1}, t_{i+1}, \ldots, t_{n-1})$, where $(t_1, \ldots, t_{n-1})$ is the solution of the second equation. We observe that the solutions that we get by solving the first and second sets of equations must correspond to the minima, and we can easily show that this maps to the minima of $L$ by setting $t_{i+1} = t_i$ or $t_i = t_{i+1}$. Since there is a unique minima of $L$, $\bar{t}' = \bar{t}''$.

Let us now see the constraints that are obtained by equation $\mathcal{E}_j^2$, where $j \neq i+1$. In our solution $t'$, we have that $t_i = t_{i+1}$. But $\mathcal{E}_j^2$ is identical to what we obtain when we set $t_i = t_{i+1}$ in $\mathcal{E}_j$. Hence, if we start with a solution $t''$ of $\mathcal{E}_j^2$, it will also be a solution to $\mathcal{E}_j$.

Putting these together, we obtain that $t'$ satisfies all the equations for our original set of equations. $\square$

We now show that an analogous result to Lemma D.1 holds when we set $t_i = 0$ and $t_j = T$, where $j > i$. Note that at the beginning we can set either $t_1 = 0$ or $t_{n-1} = T$. Observe that this is the most general thing to do, since if we set $t_i = 0$, where $i > 1$, then by the constraints, $t_j = 0 \ \forall j < i$, we will have $t_1 = 0$. Thus, the added constraint still encompasses this solution. We now show that as $L$

has a unique optimum, setting $t_1 = 0$ leads to $t_{n-1} = T$. We capture this in general by the following lemma.

**Lemma D.2.** *If the optimal solution exists in the subspace $\mathcal{A}$ of $\mathcal{D}$, obtained by setting $t_1, t_2, \ldots, t_i$ to zero and $t_{n-j}, \ldots, t_{n-1}$ to zero. Then, there is an optimal solution in the subspace $\mathcal{A}'$ of $\mathcal{D}$ by setting $t_1, t_2, \ldots, t_k$ to zero and $t_{n-k}, \ldots, t_{n-1}$ to zero, where $k = \max(i, j)$.*

*Proof.* We show this by contradiction. Let us assume without loss of generality that $i < j$. Assume that a solution exists in the subspace $\mathcal{A}$, but not in $\mathcal{A}'$. Then this solution has a non-zero entry among $(t_i, \cdots, t_k)$, say at the $x^{th}$ position. Let us call this solution $\bar{t} = (t_1, t_2, \ldots, t_{n-1})$, where $t_1 = \cdots = t_i = 0$ and $t_{n-j} = \cdots = t_{n-1} = 0$.

For a constant $T$, let $T - \bar{t} = (T - t_1, \ldots, T - t_n)$. We call this solution $\bar{t}' = (T - t_1, \ldots, T - t_n)$.

Observe that $L(\bar{t}') = L(\bar{t})$. Therefore, if $\bar{t}$ is a global minimum to $L$, then $\bar{t}'$ is also a global minimum to $L$. But this contradicts the existence of unique global minima. □

Let $\mathcal{D}_i$ be the subspace obtained by setting $t_i = 0$ and $t_{n-i} = T$. Similarly, define $\mathcal{D}_{i+1}$. Then, we have the following theorem:

**Theorem D.3.** *Consider the subspaces $\mathcal{D}_1, \mathcal{D}_2, \mathcal{D}_3, \ldots, \mathcal{D}_{n/2}$. For each $i \in \{1, \ldots, n/2\}$, one of the following hold:*

1. *The solution lies in the interior of some $\mathcal{D}_j$, where $j \leq i$.*

2. *The solution lies on the boundary of $\mathcal{D}_1, \ldots, D_i$ and in $\mathcal{D}_{i+1}$.*

*Proof.* Let us prove the theorem by induction on $i$. For $i = 1$, we have demonstrated in Section 4.1 that the solution lies in $\mathcal{D}_1$. Hence, if it does not lie in the interior of $\mathcal{D}_1$, then it lies at the boundary of $\mathcal{D}_1$. In Lemma D.1, we have shown that if the minima of $L$ only satisfy equations of the form $t_i = t_{i+1}$, then our original set of equations also has a solution. But this cannot be the case as we have seen that $\mathcal{E}_j$ $(t_i \leq t_{i+1})$ has a solution. Hence, the minima for $L$ must also satisfy some equations of the form $t_j = 0$ or/and $t_k = T$ for some $j, k > 1$. However, if it satisfies some equation of the form $t_j = 0$ for some $j > i$, then by Lemma D.2 it also satisfies $t_2 = 0$, and $t_{n-2} = T$ and so on.

Now, let us assume that the theorem holds up to $i - 1$ and demonstrate it for $i$. By the induction hypothesis, we know that either the solution lies in the interior of $\mathcal{D}_1, \ldots, \mathcal{D}_{i-1}$ or it lies in $\mathcal{D}_i$. Let us consider the case where it lies in $\mathcal{D}_i$. If it does not lie in the interior of $\mathcal{D}_i$, then it lies on some boundary of $\mathcal{D}_i$. If the solution lies at the boundary of $\mathcal{D}_i$, then it must satisfy some of the equations that form the boundary. As the minima lies in $\mathcal{D}_i$, it satisfies $t_i = 0$ and $t_{n-i} = 0$, using Lemma D.2.

Again, by Lemma D.1, we know that if the minima to $L$ only satisfy equations of the form $t_i = t_{i+1}$ and not any other equation, then our original set of equations has a solution. Hence, the minima of $L$ must satisfy some equations of the form $t_j = 0$ or/and $t_k = T$ for some $j, k > i$. However, if it satisfies the equation of the form $t_j = 0$ for some $j > i$, it also satisfies $t_{i+1} = 0$. Also using Lemma D.2, it must satisfy $t_{n-i-1} = T$. Similarly by Lemma D.2 if for some $j > i$, our equation satisfies $t_{n-i-1} = T$, then it also satisfies, $t_{i+1} = 0$.

Therefore, by these inequalities, the minima must satisfy $t_{i+1} = 0$ and $t_{n-i-1} = T$. This proves our lemma. □

In the next section, we show how the $T_i$'s depend on one another, similar to Section 4.2, where we expressed each $T_i$ in terms of $T_1$.

## D.2 DEPENDENCY OF $T_i$

In this section, we show how the $T_i$'s relate to one another under the new set of equations. While many of the proofs mirror those in Section 4.2, we now extend them to a more general setting by incorporating the constraint $t_a = 0$ for different values of $a$.

As we have already seen, if the solution of $L$ does not lie in the interior of $\mathcal{D}$, then it must lie in the subspace with the added constraints $t_a = 0$ and $t_{n-a} = T$ for some $a \in \{1, 2, \ldots, n/2\}$. To this end, we show how $T_i$ can be expressed in terms of $T_a$.

**Lemma D.4.** *Assuming $t_0 = \cdots = t_{a-1} = 0$, we can write $T_{a+i}$ in terms of $T_a$ as*

$$T_{a+i} = \frac{1}{a} \cdot \frac{(aT_a)^{i+1}}{(aT_a + 1)^i}. \tag{13}$$

*Proof.* Using Lemma 4.3, we have

$$T_i = \frac{-1 + \sqrt{1 + 4H_{i-1}H_{i-2}T_i^2}}{2H_{i-1}}$$

As a base case, when $i = 0$ we have, $\frac{1}{a} \cdot \frac{(aT_a)^1}{(aT_a+1)^0} = T_a$.

As the first $a$ terms are zero, using the induction hypothesis, we now calculate $H_{a+k}$, which is the sum of the first $a + k + 1$ terms.

$$
\begin{aligned}
H_{a+k} &= \sum_{j=0}^{k} \frac{1}{T_j} \\
&= a + a \sum_{j=0}^{k} \frac{(1 + aT_1)^j}{(aT_1)^{j+1}} && \text{(Using induction hypothesis)} \\
&= a + a \sum_{j=1}^{k+1} \frac{(1 + aT_1)^{j-1}}{(aT_1)^j} \\
&= a + \frac{1}{T_1} \cdot \frac{\left(\frac{1+aT_1}{aT_1}\right)^{k+1} - 1}{\frac{1+aT_1}{aT_1} - 1} && \text{(Using geometric progression formula)} \\
&= a + \frac{1}{T_1} \cdot \frac{\left(\frac{1+aT_1}{aT_1}\right)^{k+1} - 1}{\frac{1}{aT_1}} \\
&= a + a \left(\frac{1 + aT_a}{aT_1}\right)^{k+1} - a \\
&= a \left(\frac{1 + aT_a}{aT_a}\right)^{k+1}
\end{aligned}
$$

Now, we have

$$
\begin{aligned}
T_{a+i} &= \frac{-1 + \sqrt{1 + 4H_{a+i-2}H_{a+i-1}(T_{a+i-1}^2)}}{2H_{a+i-1}} \\
&= \frac{-1 + \sqrt{1 + 4 \cdot a\left(\frac{(1+aT_1)}{aT_1}\right)^{i-1} \cdot a\left(\frac{(1+aT_1)}{aT_1}\right)^i \cdot \left(\frac{1}{a}\frac{(aT_1)^i}{(1+aT_1)^{i-1}}\right)^2}}{2a\left(\frac{1+aT_1}{aT_1}\right)^i} \\
&= (aT_1)^i \cdot \frac{-1 + \sqrt{1 + 4(aT_1 + 1)(aT_1)}}{2a(1 + aT_1)^i} \\
&= (aT_1)^i \cdot \frac{-1 + 2aT_1 + 1}{2a(1 + aT_1)^i} \\
&= \frac{1}{a} \frac{(aT_1)^{i+1}}{(1 + aT_1)^i}.
\end{aligned}
$$

$\square$

Next, similar to Lemma 4.5, we show how to express $T_{n-a+1}$ in terms of $T_a$.

**Lemma D.5.** *Assuming $t_0 = \cdots = t_{a-1} = 0$, we have*

$$T_{n-a+1} = \frac{1}{a^2} \frac{(aT_a)^{n-2a+2}}{(1+aT_a)^{n-2a}}. \tag{14}$$

*Proof.* Similar to how we obtained Equation 11, we use $\partial L / \partial t_a = 0$, and the fact that $t_0 = \cdots = t_{a-1} = 0$ and $t_{n-a+1} = \cdots = t_n = T$, we get the following

$$T_a^2 = \frac{1}{a}(T_{a+1} + \cdots + T_{n-a} + a\delta^T)$$

Hence,

$$aT_a^2 - T_{a+1} + \cdots + T_{n-a} = a\delta^T$$

Now, using Lemma D.4 we have

$$
\begin{aligned}
T_{a+1} + T_{a+2} + \cdots + T_{n-a} &= \frac{1}{a}\left(\frac{(aT_a)^2}{(1+aT_a)} + \frac{(aT_a)^3}{(1+aT_a)^2} + \cdots + \frac{(aT_a)^{n-2a+1}}{(1+aT_a)^{n-2a}}\right) \\
&= \frac{1}{a}\left(\frac{(aT_a)^2}{(1+aT_a)} \cdot \frac{1 - (\frac{aT_a}{1+aT_a})^{n-2a}}{1 - \frac{aT_a}{1+aT_a}}\right) \\
&= a(T_a)^2 - \frac{1}{a}\frac{(aT_a)^{n-2a+2}}{(1+aT_a)^{n-2a}}
\end{aligned}
$$

Thus, putting everything together, we have

$$\frac{(aT_a)^{n-2a+2}}{(1+aT_a)^{n-2a}} = a^2\delta^T. \tag{15}$$

$\square$

**Corollary D.6.** *Assuming $t_0 = \cdots = t_{a-1} = 0$, we have*

$$\delta^T = \frac{1}{a^2} \frac{(a\delta^{t_a})^{n-2a+2}}{(1+a\delta^{t_a})^{n-2a}}. \tag{16}$$

Having derived the relation between $T_{a+i}$ and $T_a$, we next use them to approximate $T_a$ and provide our near-optimal algorithm.

### D.3 CHECKING MINIMA AND APPROXIMATION

In this section, we derive the two conditions used in Algorithm 1 to find the value of $a$. We then provide the fleshed-out version of Algorithm 1 in Algorithm 5 and derive our main theorem for the general setting ($a \neq 1$).

Similar to Section 4.3, we first define

$$g(T_a) = \frac{1}{a^2} \cdot \frac{(aT_a)^{n-2a+2}}{(1+aT_a)^{n-2a}} \tag{17}$$

**Lemma D.7.** *$g(T_a)$ is strictly monotonically increasing with $T_a \in [\delta^T, 1]$.*

*Proof.*

$$
\begin{aligned}
\frac{d}{dT_a} \frac{1}{a^2} \cdot \frac{(aT_a)^{n-2a+2}}{(1+T_a)^{n-2a}} &= \frac{(n-2a+2)(aT_a)^{n-2a+1}}{(1+aT_a)^{n-2a}} - (n-2a)\frac{(aT_a)^{n-2a+2}}{(1+aT_a)^{n-2a+1}} \\
&= \frac{(n-2a+2)(aT_a)^{n-2a+1} + 2(aT_a)^{n-2a+2}}{(1+aT_a)^{n-1}} \\
&> 0
\end{aligned}
$$

$\square$

**Corollary D.8.** $g(T_a) = \delta^T$ *admits at most one solution.*

We have shown in Lemma D.5 that any solution that satisfies Equation 15 is the optimum solution. To check if Equation 15 admits a solution, it suffices to check the value of $g(T_a)$ at $t_a = 0$ and $t_a = T$. Precisely, we need to check if

- at $t_a = 0$

$$\frac{a^{n-2a}}{(a+1)^{n-2a}} \geq \delta^T$$

- at $t_a = T$

$$\frac{1}{a^2} \cdot \frac{(a\delta^T)^{n+2-2a}}{(a\delta^T + 1)^{n-2a}} \leq \delta^T$$

The two above inequalities can be verified in $O(\ln(nT))$ time. According to Theorem D.3, domain reduction, which involves checking whether the inequalities hold and incrementing $a$ if they do not, needs to be performed at most $n/2$ times. If we find that a feasible solution to these inequalities appears after performing domain reduction up to $a - 1$ times (for some $a \leq n/2$), then, by Theorem D.3, this solution lies in the interior of the domain $D_a$. If we have to do domain reduction $n/2$ times, then there is only one point in the feasible region i.e., $t_i = 0$ for $i \leq n/2$ and $t_i = T$ for $i > n/2$ (if $n$ is odd, then $t_{n/2} = T/2$).

Once we find a value of $a$ that satisfies both the conditions, we perform binary search using Algorithm 5, which runs in $O\left((3n + T\log_2(1/\delta))\ln(nT)\right)$ time. This yields an approximate solution to $T_a$ with accuracy $1/8^n$, following an analysis similar to that in Lemma C.4. As in Lemma C.3, we can show that a solution to $g(T_a) = \delta^T$ corresponds to a solution to $\nabla L = 0$, and thus a minimum to $L$.

After computing $T_a$, we can recover the value of $T_i$'s using Lemma D.4 in $O(n\log(nT))$ time. Applying a similar analysis as in Lemma 4.6, we obtain all $T_i$ with accuracy $1/4^n$. Finally, using these approximate values of $T_i$, we compute the optimal schedule $t_i$ in an additional $O(n\ln(nT)\ln(1/\delta))$ time, with an additive error of at most $\frac{1}{2^n}$.

Note that Algorithm 5 is the complete description of Algorithm 1. We now prove one of our main theorems.

**Theorem D.9.** *From Algorithm 5, we can derive a $1/2^n$-approximate schedule to the optimal schedule using Algorithm 2.*

*Proof.* As shown in Theorem 4.2 o Section A, there exists a unique optimal schedule. In Section B, Lemmas 4.3 and 4.5 establish a system of equations whose solution yields this optimal schedule. Lemma 4.7 in Section 4.3 shows that our algorithm provides a $1/2^n$-approximation to the solution of these equations. For the general case $a \neq 1$, Lemmas D.4 and D.5 in Section D present analogous expressions, and we show that the algorithm achieves the same $1/2^n$-approximation in this broader context. Putting everything together, we conclude that our algorithm yields a $1/2^n$-approximation to the optimal schedule.

$\square$

# E MISSING DETAILS OF SECTION 5

In Section 5, we discussed many behaviors of our near-optimal solution. In this section, we prove them in detail. To begin, we first present the following property of our solution.

**Lemma E.1.** *For the optimal solution $\bar{t}$, we have $t_1 - t_0 = t_n - t_{n-1}$.*

*Proof.* At optimum, we have $T_n = \frac{T_1^n}{(1+T_1)^{n-2}}$ and $T_{n-1} = \frac{T_1^{n-1}}{(1+T_1)^{n-2}}$. Hence,

$$\frac{T_n}{T_{n-1}} = T_1 = \frac{T_1}{T_0}$$

---

**Algorithm 5** Binary Search with checking

---

1: $low \leftarrow \delta^T$
2: $high \leftarrow 1$
3: $n_{rounds} \leftarrow 3n + T \log{(1/\delta)}$
4: $mid \leftarrow (low + high)/2$
5: $curr \leftarrow 1$
6: $a \leftarrow 1$
7: $ctr \leftarrow 1$
8: **while** $ctr \leq n/2$ **do**
9:    **if** $\frac{a^{n-2a}}{(a+1)^{n-2a}} < \delta^T$ **then**
10:       $ctr++$
11:    **else if** $\frac{1}{a^2} \cdot \frac{(a\delta^T)^{n+2-2a}}{(a\delta^T+1)^{n-2a}} > \delta^T$ **then**
12:       $ctr++$
13:    **else**
14:       $a \leftarrow ctr$
15:       break
16:    **end if**
17: **end while**
18: **if** $ctr \geq n/2$ **then**
19:    **return** $(n/2, \perp)$
20: **end if**
21: $h(T_a) = \frac{1}{a^2} \cdot \frac{(aT_a)^{n-2a+2}}{(1+aT_a)^{n-2a}}$
22: **while** $curr \leq n_{rounds}$ **do**
23:    $mid \leftarrow (low + high)/2$
24:    **if** $h(mid) = 0$ **then**
25:       **return** $mid$
26:    **else if** $h(mid) < 0$ **then**
27:       $low \leftarrow mid$
28:    **else**
29:       $high \leftarrow mid$
30:    **end if**
31:    $curr \leftarrow curr + 1$
32: **end while**
33: **return** $(a, mid)$

---

The above equation can be written as

$$\delta^{t_n - t_{n-1}} = \delta^{t_1 - t_0}$$

Therefore, we get

$$t_n - t_{n-1} = t_1 - t_0$$

$\square$

Next, we show that the ratio between $T_i$ and $T_{i-1}$ is always fixed.

**Lemma E.2.** *For $i \in \{2, \ldots, n-1\}$, we have*

$$\frac{T_i}{T_{i-1}} = \frac{T_1}{(1 + T_1)}. \tag{18}$$

*Proof.* We have,

$$\frac{T_i}{T_{i-1}} = \frac{\frac{T_1^i}{(1+T_1)^{i-1}}}{\frac{T_1^{i-1}}{(1+T_1)^{i-2}}} \qquad \text{Using Theorem 4.4}$$
$$= \frac{T_1}{(1 + T_1)}$$

$\square$

**Corollary E.3.** $t_2 - t_1 = t_3 - t_2 = \cdots = t_{n-1} - t_{n-2}$.

Now, in the following two lemmas, we show that as we increase the value of $\delta$, the value of $t_1$ decreases. We first show that $t_1$ cannot be greater than $T/n$.

**Lemma E.4.** *Given $\delta, T$ and $n$, in any optimal solution $\bar{t}$ we have $t_1 < T/n$ .*

*Proof.* From Corollary B.4, we know that when $\nabla L = 0$, we have

$$\frac{\delta^{t_1 n}}{(1 + \delta^{t_1})^{n-2}} = \delta^T \tag{19}$$

Let us define $h(t_1) = \frac{\delta^{t_1 n}}{(1+\delta^{t_1})^{n-2}}$. At optimum, we would like $h(t_1) = \delta^T$.

For a fixed $\delta$,

$$h(T/n) = \frac{\delta^{n \frac{T}{n}}}{(1 + \delta^{\frac{T}{n}})^{n-2}}$$
$$< \delta^T$$

For a fixed $\delta, T$ and $n$, we know that $g(T_1) = \frac{T_1^n}{(1+T_1)^{n-2}}$ is strictly monotonically decreasing with $T_1$ (using Lemma C.1).

We have that $h(t_1) = g(\delta^{t_1})$. Since $\delta^{t_1}$ monotonically decreases with $t_1$, we obtain that $h(t_1)$ monotonically decreases with $t_1$.

We know that $h(T/n) < \delta^T$. Since in the optimal solution, we require $h(t_1) = \delta^T$, we obtain $t_1 < T/n$. $\square$

**Lemma E.5.** *Fix $n$ and $T$, then as $\delta$ increases, the value of $t_1$ decreases.*

*Proof.* We know from Corollary B.4, that at optimal value of $\bar{t}$, we have

$$\delta^T = \frac{\delta^{nt_1}}{(1+\delta^{t_1})^{n-2}}$$

Differentiating the above equation from both sides, we get

$$\frac{T}{\delta}\frac{\delta^{nt_1}}{(1+\delta^{t_1})^{n-2}} = \frac{(1+\delta^{t_1})^{n-2}\left(nt_1\delta^{nt_1-1} + n\delta^{nt_1}\ln(\delta)\frac{\partial t_1}{\partial \delta}\right)}{(1+\delta^{t_1})^{2n-4}}$$
$$- \frac{\delta^{nt_1}\left((n-2)(1+\delta^{t_1})^{n-3}t_1\delta^{t_1-1} + (n-2)(1+\delta^{t_1})^{n-3}\delta^{t_1}\ln(\delta)\frac{\partial t_1}{\partial \delta}\right)}{(1+\delta^{t_1})^{2n-4}}$$

This implies that

$$T(1+\delta^{t_1}) = (1+\delta^{t_1})(nt_1 + n\delta\ln(\delta)\frac{\partial t_1}{\partial \delta}) - \delta((n-2)t_1\delta^{t_1-1} + (n-2)\delta^t\ln(\delta)\frac{\partial t_1}{\partial \delta})$$

Hence, we have

$$T(1+\delta^{t_1}) = \frac{\partial t_1}{\partial \delta}((1+\delta^{t_1})n\delta\ln\delta - (n-2)\delta^{t+1}\ln(\delta)) + nt_1 + 2t_1\delta^{t_1}$$

Therefore,

$$(\frac{\partial t_1}{\partial \delta})(\delta\ln\delta)(n+2\delta^{t_1}) = T(1+\delta^{t_1}) - (n+2\delta^{t_1})t_1$$

Rearranging the terms, we get

$$(\frac{\partial t_1}{\partial \delta})(\delta\ln\delta)(n+2\delta^{t_1}) = \delta^{t_1}(T-2t_1) + (T-nt_1)$$

Finally, we obtain

$$\frac{\partial t_1}{\partial \delta} = \frac{\delta^{t_1}(T-2t_1) + (T-nt_1)}{\delta\ln\delta(n+2\delta^{t_1})}$$

As $n > 2$ and $t_1 < T/n$, the numerator in the above equation is strictly positive. Note that $\ln(\delta) < 0$ and therefore the denominator is strictly negative. Hence, the derivative must also be strictly less than 0. This implies that $t_1$ is monotonically decreasing with $\delta$. □

Next, we show that as $\delta \to 0$, $t_i$'s are equally spaced in $T$.

**Lemma E.6.** *As $\delta \to 0$, we have $t_i \to Ti/n$.*

*Proof.* As $T_n = \delta^T$, for the optimum solution, the corresponding value of $T_1$ satisfies

$$\delta^T = \frac{T_1^n}{(1+T_1)^{n-2}}$$

But if the solution is not optimal, the above equation might not be true. Hence, we try to estimate the value of the quantity:

$$\frac{1}{\delta^T}\frac{T_1^n}{(1+T_1)^{n-2}}$$

We would like to show that as $\delta \to 0$, the value of the above equation at $T_1 = \delta^{T/n}$ converges to 1, which will imply that $\frac{T_1^n}{(1+T_1)^{n-2}} \to \delta^T$. We now show that this is true.

$$\lim_{\delta\to 0}\frac{T_1^n}{\delta^T(1+T_1)^{n-2}} = \lim_{\delta\to 0}\frac{\delta^T}{\delta^T(1+\delta^T)^{n-2}}$$
$$= \lim_{\delta\to 0}\frac{1}{(1+\delta^T)^{n-2}}$$
$$= 1$$

As we have already seen that $t_i$'s are equally spaced between $t_1$ and $t_{n-1}$ (Corollary E.3) and since $t_1 \to T/n$ and $t_{n-1} \to T-T/n$ (using Lemma E.1), as $\delta \to 0$, this gives $t_i \to iT/n$. □

The properties that we have shown so far in this section holds when $T > n \log_{1/\delta}(2)$ (i.e., $a = 1$). We now show that similar properties hold when $T \leq n \log_{1/\delta}(2)$ (i.e., $a \neq 1$). From Section D, we observe that as $\delta$ increases, $t_{a-1} = 0$ and $t_{n-a+1} = T$ for $a \in [n/2 + 1]$ hold. We now, in the next three lemmas, argue that as $\delta$ increases, $t_a$ decreases for $a \neq 1$.

Firstly, from the analysis in Section D we observe that as $\delta$ increases, the constraint $t_a = 0$ holds for larger and larger $a$. When $t_{a-1} = 0$ and $t_a \neq 0$, from Corollary 16, we can write $\delta^T = \frac{1}{a^2} \frac{(a\delta^{t_a})^{n-2a+2}}{(1+a\delta^{t_a})^{n-2a}}$. We now try to find the ranges of $\delta$ for which $\delta^T = \frac{1}{a^2} \frac{(a\delta^{t_a})^{n-2a+2}}{(1+a\delta^{t_a})^{n-2a}}$ holds.

**Lemma E.7.** *The equation*

$$\delta^T = \frac{1}{a^2} \frac{(a\delta^{t_a})^{n-2a+2}}{(1+a\delta^{t_a})^{n-2a}} \tag{20}$$

*holds in the domain* $\delta \in \left[ \left( \frac{1}{(1+\frac{1}{a-1})^{n-2a+2}} \right)^{1/T}, \left( \frac{1}{(1+\frac{1}{a})^{n-2a}} \right)^{1/T} \right]$

*Proof.* Equation 20 holds when

- Solving Equation 20 gives a positive solution for $t_a$.

- Solving the equation

$$\delta^T = \frac{1}{(a-1)^2} \frac{((a-1)\delta^{t_a})^{n-2a+4}}{(1+(a-1)\delta^{t_{a-1}})^{n-2a+2}} \tag{21}$$

 gives a negative solution for $t_{a-1}$, i.e., the equation is invalid.

Let us define $h_a(t_a) = \frac{1}{a^2} \frac{(a\delta^{t_a})^{n-2a+2}}{(1+a\delta^{t_a})^{n-2a}}$.

Using Lemma D.7, we know that $g(T_a) = \frac{1}{a^2} \cdot \frac{(aT_a)^{n-2a+2}}{(1+aT_a)^{n-2a}}$ is monotonically increasing. with $T_a$

We know that $h_a(t_a) = g(\delta^{t_a})$. Since $\delta^{t_a}$ monotonically decreases with $t_a$, we have that $h_a(t_a)$ monotonically decreases with $t_a$.

Substituting $t_a = 0$, we get

$$h(0) = \frac{1}{a^2} \frac{a^{n-2a+2}}{(1+a)^{n-2a}}$$

We know that $h(t_a) \leq h(0)$ since $t_a \geq 0$ by monotonicity. We also know that if Equation 20 has a solution $t_a$, we would have $h(t_a) = \delta^T$. Using these, we obtain that:

$$\delta^T \leq \frac{1}{a^2} \frac{a^{n-2a+2}}{(1+a)^{n-2a}}$$

$$\delta \leq \left( \frac{1}{(1+1/a)^{n-2a}} \right)^{1/T}$$

Hence, Equation 20 is valid only for $\delta \leq \left( \frac{1}{(1+1/a)^{n-2a}} \right)^{1/T}$.

By a similar argument, we can show that Equation 21 is valid only for $\delta \leq \left( \frac{1}{(1+\frac{1}{a-1})^{n-2a+2}} \right)^{1/T}$.

For $\delta \geq \left( \frac{1}{(1+\frac{1}{a-1})^{n-2a+2}} \right)^{1/T}$, we would obtain $t_{a-1} = 0$ and Equation 21 would no longer hold.

Therefore, it becomes valid to now write Equation 20 as long as $\delta \leq \left( \frac{1}{(1+1/a)^{n-2a}} \right)^{1/T}$.

In the domain $\delta \in \left[ \left( \frac{1}{(1 + \frac{1}{a-1})^{n-2a+2}} \right)^{1/T}, \left( \frac{1}{(1 + \frac{1}{a})^{n-2a}} \right)^{1/T} \right]$, we have that:

- The equation $\delta^T = \frac{1}{(a-1)^2} \frac{((a-1)\delta^{t_a})^{n-2a+4}}{(1 + (a-1)\delta^{t_{a-1}})^{n-2a+2}}$ is invalid.

- The equation $\delta^T = \frac{1}{a^2} \frac{(a\delta^{t_a})^{n-2a+2}}{(1 + a\delta^{t_a})^{n-2a}}$ is valid.

Hence, the domain in which Equation 20 holds is $\delta \in$
$\left[ \left( \frac{1}{(1 + \frac{1}{a-1})^{n-2a+2}} \right)^{1/T}, \left( \frac{1}{(1 + \frac{1}{a})^{n-2a}} \right)^{1/T} \right]$.                                                      $\square$

**Lemma E.8.** *When* $\delta = \left( \frac{1}{(1 + \frac{1}{a-1})^{n-2a+2}} \right)^{1/T}$, *we have* $t_a = T/(n - 2a)$ *as the only solution*

*to Equation 20, i.e.* $\delta^T = \frac{1}{a^2} \frac{(a\delta^{t_a})^{n-2a+2}}{(1 + a\delta^{t_a})^{n-2a}}$ .

*Proof.* We can verify the statement of the lemma by substituting $\delta = \left( \frac{1}{(1 + \frac{1}{a-1})^{n-2a+2}} \right)^{1/T}$, and $t_a = T/(n - 2a)$ into both sides of the equation.

The LHS becomes

$$\delta^T = \left( \left( \frac{1}{(1 + \frac{1}{a-1})^{n-2a}} \right)^{1/T} \right)^T$$
$$= \frac{1}{(1 + \frac{1}{a-1})^{n-2a+2}}$$
$$= (\frac{a - 1}{a})^{n-2a+2}.$$

The RHS becomes:

$$\frac{1}{a^2} \frac{(a\delta^{t_a})^{n-2a+2}}{(1 + a\delta^{t_a})^{n-2a}} = \frac{1}{a^2} \frac{a^{n-2a+2}(\frac{a-1}{a})^{n-2a+2}}{(1 + a - 1)^{n-2a}}$$
$$= \frac{1}{a^2} \frac{(a-1)^{n-2a+2}}{a^{n-2a}}$$
$$= (\frac{a - 1}{a})^{n-2a+2}.$$

As we know that $h_a(t_a) = \frac{1}{a^2} \frac{(a\delta^{t_a})^{n-2a+2}}{(1+a\delta^{t_a})^{n-2a}}$ is monotonically decreasing with $t_a$. Hence, $t_a = T/(n - 2a)$ is the only solution to Equation 20.                                                      $\square$

**Lemma E.9.** *For* $\delta \in \left[ \left( \frac{1}{(1 + \frac{1}{a-1})^{n-2a+2}} \right)^{1/T}, \left( \frac{1}{(1 + \frac{1}{a})^{n-2a}} \right)^{1/T} \right]$, *the value of* $t_a$ *which satisfies Equation 20 strictly monotonically decreases with* $\delta$ *from* $\frac{T}{n-2a+2}$ *to 0.*

*Proof.* We know that when Equation 20 holds, we have,

$$\delta^T = \frac{1}{a^2} \frac{(a\delta^{t_a})^{n-2a+2}}{(1 + a\delta^{t_a})^{n-2a}}$$

Differentiating with respect to $\delta$, we obtain that

$$\frac{T}{a^2\delta}\frac{(a\delta^{t-a})^{n-2a+2}}{(1+a\delta^{t_a})^{n-2a}} = \frac{1}{a^2}\frac{(1+a\delta^{t_a})^{n-2a}(n-2a+2)(a\delta^{t_a})^{n-2a+1}a(\delta^{t_a}\ln\delta\frac{\partial t_a}{\partial\delta}+t_a\delta^{t_a-1})}{(1+a\delta^{t_a})^{2n-4a}}$$
$$-\frac{1}{a^2}\frac{(a\delta^{t_a})^{n-2a+2}(n-2a)(1+a\delta^{t_a})^{n-2a-1}a(\delta^{t_a}\ln\delta\frac{\partial t_a}{\partial\delta}+t_a\delta^{t_a-1})}{(1+a\delta^{t_a})^{2n-4a}}$$

Hence,

$$\frac{T}{\delta}(1+a\delta^{t_a})(a\delta^{t_a}) = [(1+a\delta^{t_a})(n-2a+2)a - (a\delta^{t_a})(n-2a)a]\delta^{t_a-1}(\delta\ln\delta\frac{\partial t_a}{\partial\delta}+t_1)$$

Rearranging the terms, we get

$$T(1+a\delta^{t_a}) = [(n-2a)+2(1+a\delta^{t_a})][\delta\ln\delta\frac{\partial t_a}{\partial\delta}+t_a]$$

Therefore, separating variables, we get

$$\delta\ln\delta\frac{\partial t_a}{\partial\delta} = [T-(n-2a+2)t_a]+(a\delta^{t_a})[T-2t_a]$$

Finally, we can write $\partial t_a/\partial\delta$ as:

$$\frac{\partial t_a}{\partial\delta} = \frac{[T-(n-2a+2)t_a]+(a\delta^{t_a})[T-2t_a]}{\delta\ln\delta}$$

When $\delta = \left(\frac{1}{(1+\frac{1}{a-1})^{n-2a+2}}\right)^{1/T}$, we have for the optimal $\bar{t}$, by Lemma E.8, we have that $t_a = \frac{T}{n-2a+2}$. Observe that at this point, the derivative of $T$ with respect to $\delta$ is negative, and it remains negative as long as $t_a \leq T/(n-2a+2)$. Hence, in the domain $\delta \in \left[\left(\frac{1}{(1+\frac{1}{a-1})^{n-2a+2}}\right)^{1/T}, \left(\frac{1}{(1+\frac{1}{a})^{n-2a}}\right)^{1/T}\right]$, we have that $t_a$ decreases strictly monotonically with respect to $\delta$. $\qquad\square$

We now prove a lemma similar to Corollary E.3.

**Lemma E.10.** *At the minima, for the first non-zero and last non-T time we have that the ads are equispaced, i.e., $t_a - t_{a-1} = t_a - 0 = T - t_{n-a} = t_{n-a+1} - t_{n-a}$.*

*Proof.* From equation 14, we obtain that

$$T_{n-a+1} = \delta^T = \frac{1}{a^2}\frac{(aT_a)^{n-2a+2}}{(1+aT_a)^{n-2a}}$$

From equation 13, we obtain that

$$T_{a+i} = \frac{1}{a}\cdot\frac{(aT_a)^{i+1}}{(aT_a+1)^i}$$

Substituting $i = n - 2a$, we get

$$T_{n-a} = \frac{1}{a}\cdot\frac{(aT_a)^{n-2a+1}}{(aT_a+1)^{n-2a}}$$

Taking the ratio, we obtain

$$\frac{T_{n-a+1}}{T_{n-a}} = T_a = \frac{T_a}{T_{a-1}}$$

This implies

$$\delta^{t_{n-a+1}-t_{n-a}} = \delta^{t_a-t_{a-1}}$$

Using the fact that $t_{n-a+1} = T$ and $t_{a-1} = 0$, we have

$$\delta^{T-t_{n-a}} = \delta^{t_a-0}$$

Taking $\log$ on both sides, we get

$$T - t_{n-a} = t_a - 0$$

This proves our lemma. $\qquad\square$

Now, similar to Lemma E.2, we have the following lemma.

**Lemma E.11.**
$$\frac{T_i}{T_{i-1}} = \frac{aT_1}{aT_1 + 1}$$

*Proof.* From equation 13, we obtain that

$$T_{a+i} = \frac{1}{a} \cdot \frac{(aT_a)^{i+1}}{(aT_a + 1)^i}$$

$$\frac{T_{a+i+1}}{T_{a+i}} = \frac{1}{a} \cdot \frac{(aT_a)^{i+2}}{(aT_a + 1)^{i+1}} \times a \cdot \frac{(aT_a + 1)^i}{(aT_a)^{i+1}}$$
$$= \frac{aT_a}{aT_a + 1}$$

$\square$

Hence, the ratio between successive terms is some constant. Now we prove a lemma similar to Lemma E.5.

**Lemma E.12.** *The value of $t_a$ decreases as $\delta$ increases.*

*Proof.* From equation 14, we obtain that

$$T_{n-a+1} = \frac{1}{a^2} \frac{(aT_a)^{n-2a+2}}{(1 + aT_a)^{n-2a}}$$

We have already established that the RHS is a monotonically increasing function in Lemma D.7. Observe that $T_{n-a+1} = \delta^T$, therefore as $\delta$ increases, $T_{n-a+1}$ increases. This means that the LHS increases with delta. Since the RHS is monotonic, $T_a$ also increases with $\delta$. As $T_a = \delta^{t_a}$, $t_a$ decreases as $\delta$ increases. $\square$

We have now established all the behaviors of our near-optimal solution as described below.

**Observation E.13.** *The near-optimal ad schedule exhibits the following patterns as $\delta$ varies:*

- *As $\delta \to 0$, ads are placed at evenly spaced intervals.*

- *As $\delta$ increases, more ads concentrate at times $0$ and $T$.*

- *As $\delta$ increases, the first $t_i > 0$ moves towards 0, and the last $t_j < T$ moves towards $T$. The remaining ads are evenly spaced between $t_i$ and $t_j$.*

So far, our analysis has assumed that ads are instantaneous. We now extend our approach to the setting where all ads have equal size and show that a near-optimal schedule can still be achieved.

### E.1 ADVERTISEMENTS WITH EQUAL SIZES

Let us dive into the case where the ads are of equal size, say $s$. We want that at any point in time, there is only one ad schedule. For this problem, we consider a new loss function which is defined as

$$L_s(\bar{t}) = \sum_{j < i} \delta^{t_i - t_j - s} \tag{22}$$

The above loss function can be written as

$$L_s(\bar{t}) = \delta^{-s} \sum_{j < i} \delta^{t_i - t_j} \tag{23}$$

Note that the loss function of Equation 23 is similar to $L(\bar{t})$. Hence, the minima obtained for $L(\bar{t})$ will also be minima for $L_s(\bar{t})$. However, for feasibility, we need that the ads are at least $s$ time apart. Observe that for Equation 23, we need an interior point solution, and hence we work under the assumption that $T > n \log_{1/\delta}(2)$ (i.e., $a = 1$).

**Lemma E.14.** *When all ads are equal in length, if the second ad can be scheduled as per our near-optimal schedule, without overlapping with the first ad, then all ads can be scheduled as per our near-optimal schedule.*

*Proof.* To construct the schedule, we place our first ad at $t_0 = 0$ and the last ad at $t_n = T - s$. We then solve the problem for $T' = T - s$.

We now want to show that if the second ad can be scheduled optimally, then all ads can be scheduled optimally. To schedule the second ad, we need that $t_1 - t_0 \geq s$, which implies $t_1 \geq s$ and hence, $T_1 = \delta^{t_1} \leq \delta^s$. This immediately ensures that the second last ad i.e., $t_{n-1}$ can be scheduled as $t_1 - t_0 = t_n - t_{n-1}$ (using Lemma E.1).

We also know that for $i \in \{2, \ldots, n-1\}$

$$\frac{T_i}{T_{i-1}} = \frac{T_1}{1 + T_1}$$
$$\leq T_1 \qquad \text{(Using Lemma E.2)}$$

Therefore,

$$\delta^{t_i - t_{i-1}} \leq T_1$$
$$\leq \delta^s$$

Taking $\log$ to the base $\delta$ on both sides, we obtain

$$t_i - t_{i+1} \geq s$$

Hence, there is enough space for ads to be scheduled optimally. □

So far, we have assumed that the number of ads $n$ is known. In the next section, we show how to determine the optimal number of ads for a user.

### E.2 OPTIMIZING THE NUMBER OF ADS TO DISPLAY

Given that the relative strengths of mere exposure and operant conditioning are known, a natural question arises: how many ads should be shown to maximize the overall reward? To answer this, we observe that for any fixed number of ads $n$, the loss due to operant conditioning can be computed exactly. In addition, $\sum B(i)$ can be calculated since it is just a function of the number of pulls, allowing us to calculate the total reward $R(\bar{t})$ precisely.

To find the optimal number of ads, we evaluate $R(\bar{t})$ for each $n \in \{1, \ldots, \tilde{n}\}$, where $\tilde{n}$ is the maximum number of ads that can fit within the time horizon $T$. The value of $n$ that maximizes $R(\bar{t})$ gives the optimal number of ads. Algorithm 3 performs this procedure and returns both the optimal number of ads and a corresponding near-optimal schedule in quasi-quadratic time with respect to $\tilde{n}$. Note that in most real-world scenarios, the maximum number of ads is either known in advance or can be learned efficiently. Moreover, prior work (RedCircle, 2023; Pandora) suggests that the maximum number of ads is typically not very large. For example, platforms like YouTube and Spotify generally display around 10 and 5 ads per hour, respectively. Hence, we can conclude that Algorithm 3 performs in quasi-linear time for all practical purposes.

## F MISSING EXPERIMENTS FROM SECTION 7

In this section, we provide additional experiments omitted from Section 7. For all the experiments, we used an 11th-generation Core i5 laptop with 8GB of RAM. The experiments took a few seconds to run. Since the experiments are designed to be very lightweight, running them on any personal computer should not be much of a problem. The complete code is available at https://anonymous.4open.science/r/Ads-that-Stick-5E13/README.md. We now provide a few more experiments for the four experiments described in Section 7.

### F.1   Variation in near optimal strategy with change in $\delta$

We present our experimental results to illustrate how our strategy evolves as the parameter $\delta$ increases from 0 to 1 for an even $n + 1$.

We observe that the pattern is largely similar to the odd case, with one key difference: for an odd number of ads, the $\frac{n+1}{2}$-th ad remains fixed at $t = T/2$, regardless of the value of $\delta$ (See Figure 1a). But for an even number of ads (See Figure 2a), the optimal times to show the ads in the first half shift towards 0 while the optimal time to show ads in the second half shifts towards $T$ as the value of $\delta$ increases. Hence, it supports the theoretical behavior described in Observation 5.1.

### F.2   Our strategy vs baseline strategies

We have presented the performance of our near-optimal strategy compared to three baseline strategies, namely Uniform, Corner, and Random. We now describe them in more detail.

- **Uniform:** Ads are placed uniformly over $[0, T]$, with equal spacing.

- **Corner:** Half of the ads are placed at $t = 0$, half at $t = T$.

- **Random:** The first and last ads are fixed at $t = 0$ and $t = T$, respectively, while the remaining $n - 2$ ads are distributed uniformly at random over the interval $(0, T)$.

In Section 7.2, we presented an experiment with $n + 1 = 15$ and $T = 100$, modeling a video streaming scenario starting with $\delta = 0.9$. We now present a similar experiment for a music streaming setting (Figure 2b), where a typical session lasts around 60 minutes with approximately 6 ads. This is to emulate what popular music streaming services offer in their free plan. It can be observed that our strategy performs better than all the baseline strategies.

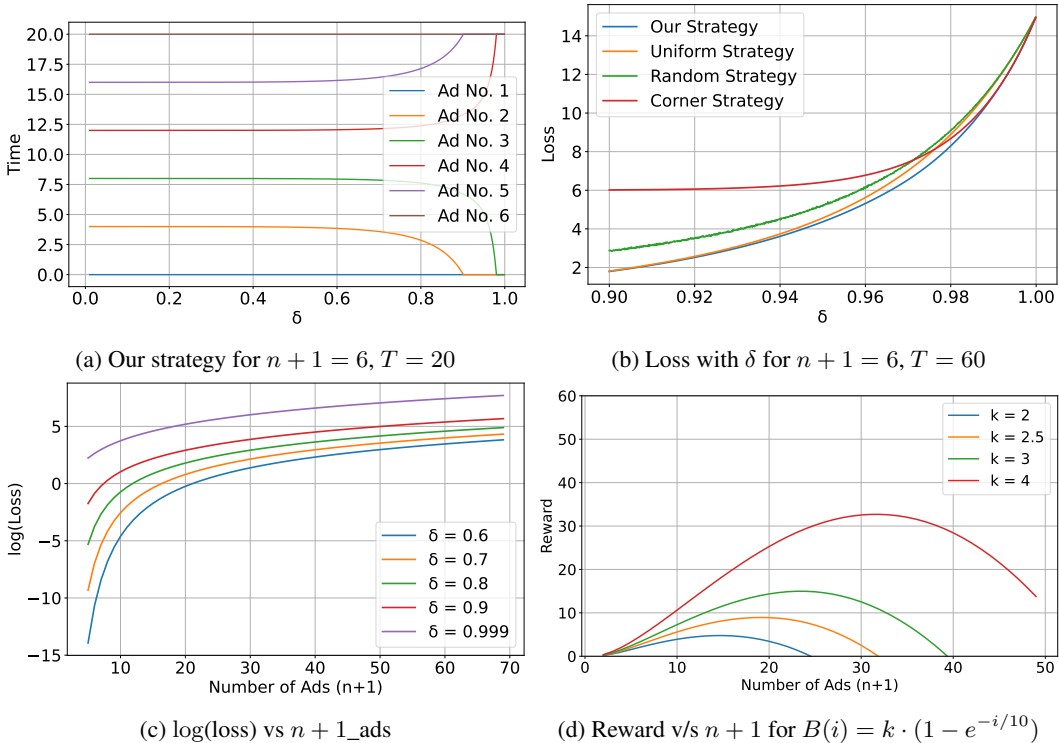

(a) Our strategy for $n + 1 = 6, T = 20$

(b) Loss with $\delta$ for $n + 1 = 6, T = 60$

(c) log(loss) vs $n + 1$_ads

(d) Reward v/s $n + 1$ for $B(i) = k \cdot (1 - e^{-i/10})$

Figure 2: Additional experiments to evaluate the performance of our near-optimal strategy.

### F.3 CHANGE IN LOSS WITH NUMBER OF ADS

In Section 7.3, we examined how the loss changes with the number of ads for different values of $\delta$. We extended that experiment by plotting $\log(\text{loss})$ versus the number of ads (Figure 2c), and observed trends similar to those in Section 7.3.

### F.4 OPTIMUM NUMBER OF ADS

In Section 7.4, we used a sigmoid reward function to determine the optimal number of ads for various values of $k$ and $c$, where $k$ captures the combined strength of hedonistic adaptation and mere exposure, and $c$ controls the sensitivity to the number of ads. We now perform a similar experiment using $B(i) = k(1 - e^{-cx})$ (See Figure 2d), and show that the peak of the curve corresponds to the optimal number of ads.

