# OpenReview forum: "Ads that Stick: Near-Optimal Ad Optimization through Psychological Behavior Models"
_ICLR.cc/2026/Conference — ICLR 2026 Poster_

### Official Review · Reviewer_tYzy · 2025-10-28

**Soundness:** 3
**Presentation:** 2
**Contribution:** 3
**Rating:** 8
**Confidence:** 4

**Summary:**

The paper performs a theoretical quantative study of a model of ad fatigue, which is important in advertising, by considering the number of ads shown to a user and the effect of ad recency on the user's inclination to positively interact with an ad. A bi-level algorithmic framework is established for computing an optimal ad schedule given a fixed number of ads as the 'inner' problem, and computing the optimal number of ads as the 'outer' problem.

Despite a few weaknesses, the work is enlightening and interesting.

**Strengths:**

1. The phenomenon is interesting and important.
2. A fresh deviation from all this "attention is all you need" trend, where a phenomenon is actually studied in depth.
3. The framework is simple, without many 'moving parts'.
4. The bi-level idea is nice and practical, if indeed we aim to plan an ad schedule for each user separately.
5. The results are insightful. For example, it's not trivial that the optimal schedule tends to "show a lot only at the beginning and the end of the time-frame" only when the fatigue parameter is very close to 1, and otherwise close-to-uniform strategy appears optimal (Figure 1 (a))

**Weaknesses:**

1. In practice, privacy concerns don't let you, in many cases, know when you're showing an ad to the same user. So such an idea is practical mainly in "closed gardens" where all users are always logged in and we know who they are. Most advertising marketplaces are not like that.
2. Code to reproduce the experiments is important, and any claim to publish it is missing from the paper.
3. The function B(i) is concave and increasing, but we don't know if it's the right model. There are many ways to model a concave and not necessarily increasing function. It may eventually asymptotically be concave and increasing, but it might be the case that the effect of its shape before asymptotic behavior kick in is not trivial.

**Questions:**

1. Will the code to reproduce the results be published?
2. Why do you believe that a concave and increasing function on the entire domain is a reasonable model to gain insights?
3. How do you propose fitting the parameters of such a model if we want to actually use it in an advertising platform?

---

> ### Author Response · Authors · 2025-11-20
>
> We thank the reviewer for their thoughtful comments and are pleased that you found our work enlightening and interesting. Below, we address the points you raised.
>
> Suitability of Strategy: You are correct that, due to privacy constraints, it may not always be possible to repeatedly identify the same user. We believe that our strategy is best suited to:
> (1) Non-personalized aggregate settings, such as TV and radio ads. Here an ad placement is not personalized but rather tailored for an average user from a target demographic using historical data.
> (2) As you rightfully mentioned, in closed-garden settings where users are consistently logged in and can be distinguished across visits.
> (3) Platform-level scheduling, after the initial bidding process.
>
> Code: The code to reproduce and validate the results are provided in Appendix F. As Reviewer Jqk7  commented, “The code provided was functional and reproduced the experimental claims”. We will definitely include the link to our open code in the main body of the paper so that it is easily accessible to anyone.
>
> Concave and increasing function: You are right that $B(i)$ may be asymptotically concave and increasing, but not necessarily so near the beginning. Tackling such functions requires a rather different form of analysis, which we intend to pursue in our future work. As described in the paragraph beginning at Line 86, the function $B$ models the positive effects of mere exposure and hedonic adaptation, and concavity is justified by the “diminishing returns” behavior inherent in these psychological phenomena. We assume that the function $B(i)$ is concave and increasing, following prior work (Patil et al., 2023; Blum & Ravichandran, 2025). Further, as mentioned in Line 212, our analysis extends to any function $B$.
>
> Fitting Parameters: Fitting the parameters of the model using experimental data is something that requires very careful analysis and precise setups to prevent spurious correlations, and is something we are thinking to do in our future work. However, one proposal could be as follows: One can observe that the solution of our model is of a very particular form and can be determined by two parameters: (i) the number of ads to show at the beginning, and (ii) the time at which to show the first non-beginning ad. Hence, one could do a grid search over these parameters for a representative subset of the target users to find the optimal parameters based on a suitable metric of user engagement. Further, in a closed garden setting, one could potentially introduce learning algorithms to learn these two parameters by evaluating user interest over each ad by various metrics.
>
> We thank you again for your review. We will definitely include the suggestions you recommended.
>
>
> References
>
> Avrim Blum and Kavya Ravichandran. Nearly-tight approximation guarantees for the improving
> multi-armed bandits problem. In Algorithmic Learning Theory, pp. 228–245. PMLR, 2025.
>
> Vishakha Patil, Vineet Nair, Ganesh Ghalme, and Arindam Khan. Mitigating disparity while
> maximizing reward: tight anytime guarantee for improving bandits. In Proceedings of the Thirty-
> Second International Joint Conference on Artificial Intelligence, pp. 4100–4108, 2023.

---

### Official Review · Reviewer_CFrE · 2025-11-01

**Soundness:** 4
**Presentation:** 3
**Contribution:** 4
**Rating:** 6
**Confidence:** 3

**Summary:**

A reward model for advertising is derived from the marketing literature.
In the context where we have to expose a user during a known interval, an optimization algorithm is derived.
Some qualitative insights are provided on the behavior of the solution.

**Strengths:**

The authors did an excellent job at introducing a reward model by bridging results from other fields.
I found the article very informative and inspiring.

**Weaknesses:**

Main weakness : the technical sections of the paper should be improved
* the graphs cannot be read properly (no color)
* discussion and explanation on the baselines in the main text would be welcome
* the proof of the statements are sometime two hand waving, the mathematical writing should be improved
* I am not sure the algorithm itself is a very technical contribution (mostly basic maths). While this is not a reason for rejection or anything, I believe the proof sketch should be better articulated

**Questions:**

* Can you provide an updated version of figure 1?
* could you provide a summary of the proof that would clearly delineate what is easy and what is hard in the derivations?

---

> ### Author Response · Authors · 2025-11-20
>
> We sincerely thank the reviewer for their thoughtful and encouraging feedback. We are glad to hear that you found our work informative and inspiring. Below, we address the points you raised.
>
> Readability of graphs: We apologize that Figure 1 was not sufficiently clear. We will update the figure to improve its readability.
>
> Baselines: Regarding the baselines, we considered three strategies: Uniform, Corner, and Random. These were placed in the appendix due to space constraints. Following your suggestion, we will move them into the main body of the paper to make the comparisons more accessible.
>
> Mathematical writing of proofs: In the main body of the paper, we present the key lemmas and theorems with limited proof details, primarily due to space limitations. This may at times make the exposition feel high-level. However, we have provided a rigorous analysis with complete proofs for all results in the appendix, which the reviewer Jqk7 also acknowledged as “Solid and Complete Theoretical Contributions.” That said, we appreciate your concern and will add additional details or clarifications in the proofs wherever needed.
>
> Summary of proofs: The reviewer Jqk7 has generously provided an overview of our main technical reviews in their review, which we would like to direct you to. Furthermore, our formal analysis of the optimal policy itself requires techniques from convex optimization and involves non-trivial calculations. It is quite true that the resulting algorithm is quite simple, and we view that as a strength of our paper. However, we feel that in order to reach this algorithm, the underlying analysis was non-trivial.
>
> We thank you again and believe that we have addressed all your concerns. We will incorporate the recommended revisions to further improve the clarity and readability of the paper.

---

> > ### Comment · Reviewer_CFrE · 2025-11-26
> >
> > Thank you for your answer, I will update my score accordingly.

---

### Official Review · Reviewer_Jqk7 · 2025-11-01

**Soundness:** 3
**Presentation:** 2
**Contribution:** 3
**Rating:** 6
**Confidence:** 3

**Summary:**

This paper proposes a principled algorithmic framework for scheduling advertisements by modeling three psychological effects: mere exposure, hedonic adaptation, and operant conditioning. The authors formulate ad scheduling as an optimization problem, specifically minimizing a loss function $L(\overline{t})$ that depends on the temporal decay of user interest.

1. A novel psychological model for ad scheduling.
2. A quasi-linear time algorithm (Algorithm 1 / Algorithm 5) that finds a near-optimal schedule with provable, exponentially small approximation error.
3. Theoretical analysis proving the uniqueness of the optimal solution (via strict convexity) and deriving a novel recursive structure for the optimal ad timings.
4. Experimental validation showing that the proposed schedule outperforms heuristics like uniform spacing by over 10% in realistic parameter regimes (e.g., high $\delta$).

Having thoroughly reviewed the main paper and all appendices (A-F), I find the work to be technically sound, novel, and practically relevant. The theoretical claims are correct. My main concerns are related to presentation, as much of the core theoretical justification is relegated to the appendices, making the main paper difficult to follow and appear incomplete.

**Strengths:**

1. **Well-Motivated Problem:** The paper addresses a practically important problem with clear economic implications. The grounding in established psychological principles (mere exposure, hedonic adaptation, operant conditioning) is a significant strength and is well-modeled.
2. **Solid and Complete Theoretical Contributions:** The main theoretical results are non-trivial and, upon review of the appendices, are fully proven.
    - **Uniqueness (Thm 4.2):** The proof of strict convexity and a unique minimum is correct and detailed in Appendix A.
    - **Recursive Structure (Thm 4.4, Lemma 4.5):** The recursive formulas for $T_i$ are proven in Appendix B. I have numerically verified these formulas.
    - **General $a \neq 1$ Case (Appendix D):** My initial concern about the analysis being limited to $a=1$ was unfounded. Appendix D provides a complete and correct analysis for the general (boundary) case $a \neq 1$, deriving the analogous recursive relations (Lemma D.4) and the general solution condition (Lemma D.5).
    - **Approximation Guarantees (Cor 4.8, Thm D.9):** The exponentially small approximation error $|t_i^* - t_i| \le 1/2^n$ is proven for the $a=1$ case in Appendix C and generalized for all $a$ in Theorem D.9.
3. **Practical and Efficient Algorithm:** The proposed algorithm (detailed in Algorithm 1 and Algorithm 5) is efficient (quasi-linear time) and implementable. The code provided was functional and reproduced the experimental claims.
4. **Validated Experimental Claims:** All experimental claims in Section 6 are verified. The results showing uniform spacing is optimal for small $\delta$ and endpoint-concentration is optimal for large $\delta$ align with the theory and provide valuable, actionable insights.

**Weaknesses:**

The paper's primary weaknesses are not technical but related to presentation and clarity. The main paper, read in isolation, is confusing and appears incomplete.

1. **Critical analysis deferred to Appendix:** The paper is not self-contained. The analysis for the general $a \neq 1$ case (Appendix D) and the proofs for all key theorems (Thm 4.2, Thm 4.4) and approximation bounds are entirely in the appendix. While this is common, the $a \neq 1$ analysis is a _core_ part of the algorithmic contribution (Algorithm 1) and its omission from the main text makes the paper's claims seem unsupported.
    - I would suggest to add theorem statements for the general $a \neq 1$ case in the main paper. Also, add 1-2 paragraph proof sketches for Theorems 4.2 and 4.4 to the main text.
2. **Algorithm presentation is opaque:** The conditions in Algorithm 1 are opaque and unmotivated. The appendix (Algorithm 5 and Appendix D.3) reveals these are the boundary-checking conditions for finding the correct $a$.
    - I would recommend to add a remark after Algorithm 1 explaining the purpose of these conditions, as detailed in Appendix D.3.
3. **Issues pertaining to confusing notation:** The use of "$n+1$ ads" with indices 0 to $n$ is non-standard and causes confusion. The relationship between the paper's $n$ and the code's $n$ is also implicit.
    - Add a notation table at the start of Section 2. Explicitly define $T_1$ in Section 4.2 and its relationship to $T_a$.
4. **Limited experimental scope:** The experiments are entirely synthetic. While they effectively validate the algorithm's properties, the paper would be stronger if it addressed how the model parameters ($\delta$, $B(\cdot)$) could be calibrated using real-world user data.
    - Having a discussion on parameter estimation and acknowledge the limitation of not having real ad-serving data would help improve the paper.

**Questions:**

My initial Questions were largely resolved by the appendix in the supplementary material.

- **Q: How sensitive are the results to misspecification of $\delta$?**
    - This question remains open. A sensitivity analysis would be a valuable addition, as $\delta$ is unlikely to be known perfectly in practice.

This paper provides solid theoretical contributions and a practical, efficient algorithm. With the appendices, it is clear the technical work is sound. The required revisions are purely focused on improving clarity and presentation to ensure the paper's contributions are properly understood by the reader.

---

> ### Author Response · Authors · 2025-11-20
>
> We sincerely thank the reviewer for their positive and encouraging feedback, and for carefully confirming the correctness of the technical part of our analysis. We are delighted to hear that you found our work novel, technically sound, and practically relevant. Below, we address the points raised in your review.
>
> Critical analysis deferred to the appendix: We agree that many technical details, including the proofs of Theorem 4.2 and Theorem 4.4, are currently placed in the appendix. These proofs are substantial and nontrivial, and therefore were included in the appendix. Nevertheless, we will add proof sketches for the main theorems in the main body of the paper to improve clarity for the readers.
> The general case $a \neq 1$ was similarly deferred due to space constraints and because its proof structure closely matches that of the $a = 1$ case, though with additional technical details. Following your suggestion, we will include the theorem and a high-level proof sketch for the general case in the main body of the paper to enhance readability.
>
> Opacity in algorithm presentation: As recommended, we will add a remark clarifying the inequalities and conditions involving the parameter $a$.
>
> Notation concerns: We appreciate the feedback and will revise the notation to eliminate any ambiguity.
>
> Limited experimental scope: Our current experiments focus on synthetic data to illustrate the correctness of our algorithm and its implications. Learning the parameter $\delta$ is definitely a great next step to improve the paper. We will definitely include this in our future work. We will also include the lack of experiments on real ad-serving data as one of our limitations in Section 7.1 to provide a better understanding of the paper's scope.
>
> Sensitivity to misspecification of $\delta$. We agree that understanding robustness to $\delta$ misspecification is an interesting question for future research. Formulating the right notion of sensitivity to analyze itself is a valid question. Further, learning $\delta$ is a meaningful open problem, and we intend to explore this in subsequent work.
>
> We again thank the reviewer for their constructive suggestions. We will incorporate them to improve the clarity and readability of the paper.

---

### Meta-Review · Area_Chair_AQ41 · 2025-12-30

**Summary:**

The reviewers agreed that the paper presents a technically sound and novel approach to ad scheduling based on psychological behavior models, with strong theoretical guarantees and insightful results. The main concerns raised were not about correctness or novelty, but about clarity and presentation, including the reliance on appendices for key proofs, limited intuition around the algorithm in the main text, clarity of notation and figures, and the use of synthetic experiments with limited discussion of real-world applicability.

**Reviewer Concerns:**

The rebuttal addresses the main reviewer's concerns regarding clarity and presentation. The authors commit to adding proof sketches and theorem statements to the main paper, clarifying the algorithmic conditions, improving notation and figures, making experimental baselines more explicit, and clearly linking the provided code. The remaining limitations relate to the lack of real-world experimental validation and parameter estimation, which the authors appropriately acknowledge as future work rather than unresolved flaws.

**Reviewer Scores:**

Based on the rebuttal, I expect a positive score adjustment from the two reviewers who raised primarily presentation-related concerns, with one reviewer explicitly stating they would update their score upward. The third reviewer was already strongly positive, and no material score change is expected in that case.

---

### Decision · Program_Chairs · 2026-01-26

Accept (Poster)